# Evolution of a fuzzy ribonucleoprotein complex in viral assembly

Huaying Zhao[1], Tiansheng Li[2], Sergio A Hassan[3], Ai Nguyen[1], Siddhartha AK Datta[1], Guofeng Zhang[4], Camden Trent[1], Agata M Czaja[1], Di Wu[5], Maria A Aronova[6], Kin Kui Lai[7], Grzegorz Piszczek[5], Richard Leapman[6], Jonathan W Yewdell[2], Peter Schuck[1]*

[1]Laboratory of Dynamics of Macromolecular Assembly, National Institute of Biomedical Imaging and Bioengineering, National Institutes of Health, Bethesda, United States; [2]Cellular Biology Section, Laboratory of Viral Diseases, National Institute of Allergy and Infectious Diseases, National Institutes of Health, Bethesda, United States; [3]Bioinformatics and Computational Biosciences Branch, National Institute of Allergy and Infectious Diseases, National Institutes of Health, Bethesda, United States; [4]Electron Microscopy Unit, Trans-NIH Shared Resource on Biomedical Engineering and Physical Science, National Institute of Biomedical Imaging and Bioengineering, National Institutes of Health, Bethesda, United States; [5]Biophysics Core Facility, National Heart, Lung, and Blood Institute, National Institutes of Health, Bethesda, United States; [6]Laboratory of Cellular Imaging and Macromolecular Biophysics, National Institute of Biomedical Imaging and Bioengineering, National Institutes of Health, Bethesda, United States; [7]HIV Dynamics and Replication Program, Center for Cancer Research, National Cancer Institute, Frederick, United States

*For correspondence:
schuckp@mail.nih.gov

Competing interest: The authors declare that no competing interests exist.

## eLife Assessment

This is a **valuable** study that combines biophysical and evolutionary approaches to understand why particular mutations in the SARS-CoV-2 protein N arose during the COVID-19 pandemic. The evidence is **solid** and supports the conclusions.

**Abstract** Previously, we showed that the genetic diversity of SARS-CoV-2 nucleocapsid (N) protein explores a wide range of biophysical properties facilitated by non-local impact of point mutations to its intrinsically disordered regions (Nguyen et al., 2024). This includes modulation of self-association, such as the creation of a de novo binding interface through the P13L mutation characteristic of Omicron variants. In the present work, we focus on the key function of N condensing viral RNA into ribonucleoprotein particles (RNPs) for viral assembly. Lacking high-resolution structural information, biochemical and biophysical approaches have revealed architectural principles of RNPs, which involve cooperative interactions of several protein-protein and protein-RNA interfaces, initiated through oligomerization of conserved transient helices in the central disordered linker of N. Here, we study the impact of defining N-protein mutations in variants of concern on RNP formation, using biophysical tools, a virus-like particle assay, and reverse genetics experiments. We find convergent evolution in repeated, independent introduction of amino acid substitutions strengthening existing binding interfaces, compensating for other substitutions that promote viral replication but decrease RNP stability. Furthermore, we show that the P13L mutation of Omicron variants enhances RNP assembly and increases viral fitness. Overall, our data reveal RNP complexes to be highly variable not only in sequence and conformations but also in thermodynamic and kinetic stability,

with its pleomorphism affecting basic architectural principles. We hypothesize that the formation of polydisperse, fuzzy N-RNA clusters with multiple distributed weak binding interfaces optimizes reversible RNA condensation, while supporting host adaptation and allowing for a large sequence space to be explored.

## Introduction

Intrinsically disordered proteins are ubiquitous and play key roles in many dynamic cellular processes, including cellular signaling, transcriptional regulation, as well as spatio-temporal organization (*Dyson and Wright, 2005*; *Holehouse and Kragelund, 2024*). Intrinsic disorder is particularly prevalent in proteins of RNA viruses, for reasons deeply connected to evolution in presence of their high mutation rates and quasispecies nature (*Brown et al., 2011*; *Gupta and Uversky, 2024*; *Tokuriki et al., 2009*; *Xue et al., 2014*).

On a physicochemical level, disorder supports multi-functionality through modulation of the conformational ensemble dependent on ligands and post-translational modifications (*Botova et al., 2024*; *Carlson et al., 2022*; *Carlson et al., 2020*; *Jack et al., 2021*; *Ranganathan et al., 2023*; *Savastano et al., 2020*; *Syed et al., 2024*) and the local environment (*Nesmelova et al., 2019*; *Roden et al., 2022*), the possibility of transient folding (*Alderson et al., 2023*; *Bessa et al., 2022*; *Zachrdla et al., 2022*; *Zhao et al., 2022*), and the propensity for protein clustering and liquid-liquid phase separation (LLPS; *Alberti et al., 2019*; *Brocca et al., 2020*; *Perdikari et al., 2020*; *Ranganathan and Shakhnovich, 2020*; *Savastano et al., 2020*; *Zachrdla et al., 2022*). Disorder also permits a significant degree of sequence variation, as many functionally important features are not dependent on the specific sequence but are encoded in non-local biophysical properties (*Alston et al., 2023*; *Nguyen et al., 2024*; *Zarin et al., 2021*). Opposite to the achievement of mutational tolerance through high stability, such as exhibited, for example, by thermostable enzymes, mutational tolerance can be achieved in loosely packed and disordered proteins by lowering the potentially deleterious effect of mutations on stability (*Tokuriki et al., 2009*; *Tokuriki and Tawfik, 2009*). In addition, disorder and flexibility magnify potential impact of mutations on conformation and biophysical properties (*Nguyen et al., 2024*). Furthermore, the high degree of sequence variability supports the exploration of short linear interaction motifs (SLiMs) in intrinsically disordered regions for interfacing with a large variety of eukaryotic regulatory and signaling processes, thus favoring adaptability and evolvability on the level of the host-virus interface (*Davey et al., 2011*; *Duro et al., 2015*; *Hagai et al., 2014*; *Schuck and Zhao, 2023*).

In recent years, it has become apparent that assemblies of disordered proteins frequently retain significant disorder and conformational flexibility of their interaction partners, generating 'fuzzy' complexes (*Longhi et al., 2017*; *Tokuriki et al., 2009*; *Tompa and Fuxreiter, 2008*). The potential functional advantages of intrinsic disorder in protein interactions include leveraging of weak binding interfaces through allovalency and the adaptability to multiple binding partners (*Longhi et al., 2017*; *Olsen et al., 2017*). In fuzzy complexes, the total binding energy is distributed into multiple distinct ultra-weak interaction sites (*Olsen et al., 2017*). Similar to individual RNA virus proteins with loose or absent structure, maintaining disorder and a spatial distribution of low-energy interactions in the protein complexes may increase the tolerance for mutations and improve evolvability of protein complexes.

The unprecedented worldwide sequencing effort of SARS-CoV-2 genomes during its rapid evolution in humans provides a unique opportunity to examine these concepts. The genomic database now exceeds in size that of any other virus by orders of magnitude (*Elbe and Buckland-Merrett, 2017*; *Rochman et al., 2022*), providing the basis for phylogenetic analyses and for monitoring the emergence of variants of concern and their geographic spread (*Hadfield et al., 2018*). In addition, it exhaustively samples the mutational landscapes of amino acids that can occupy any position of the viral proteins, which reflects their biophysical constraints (*Bloom and Neher, 2023*; *Nguyen et al., 2024*; *Zhao et al., 2022*). For example, the SARS-CoV-2 nucleocapsid (N-)protein – the most abundant viral protein in infected cells and the focus of the present work – has 419 positions, 86% of which can be assumed by on average four to five different amino acids, and up to 12 in positions in the three intrinsically disordered regions (*Zhao et al., 2022*; *Figure 1A*), evidently without fatally compromising dozens of reported N-protein functions (*Wu et al., 2023*). Observed N-protein mutations

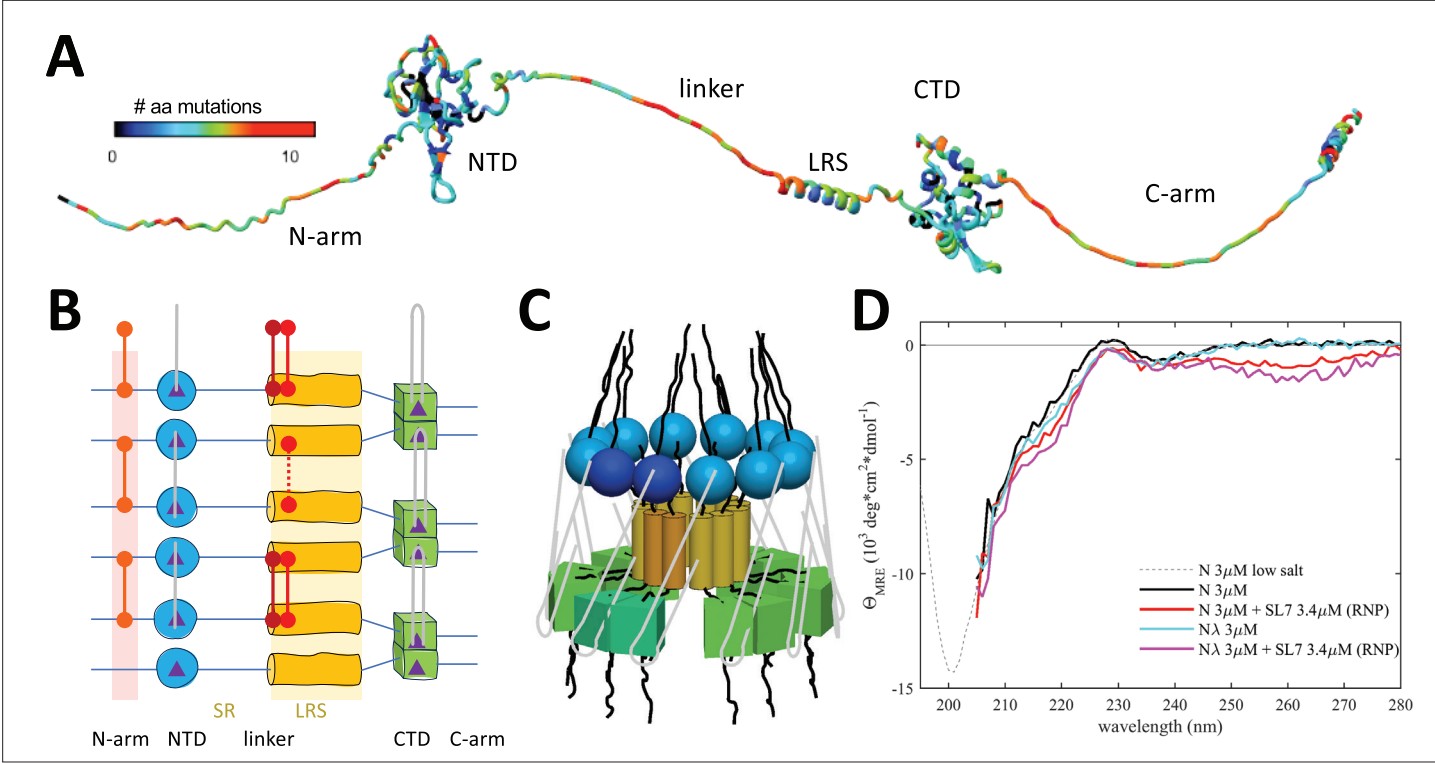

**Figure 1.** Basic organization of N-protein and RNPs. N-protein (1-419) has two folded domains, NTD (45-180) and CTD (248-363), and three intrinsically disordered regions including the N-arm (1-44), the central linker (181-247), and the C-arm (364-419). (**A**) Displayed is an AlphaFold2 structure where the disordered N-arm, linker, and C-arm are artificially stretched for clarity. The residues are color-coded according to the number of different amino acids that have been observed at this position in the mutational landscape replacing the Wuhan-Hu-1 sequence. The bioinformatic analysis was carried out as described previously (**Zhao et al., 2023**), updated to August 26, 2024, using a threshold of >5 genomes for each mutation. (**B**) Schematic of protein-protein and protein/RNA interfaces in RNP assembly. The nucleic acid binding domain at the N-terminus (NTD) is indicated in blue, the LRS in yellow, and the dimerization domain (CTD) in green. Regions of self-association are indicated by shaded backgrounds. The linker is subdivided into a serine and arginine-rich region (180–205, SR) and a L-rich region (206–247, LRS). LRS can transiently fold into helices that create a hydrophobic patch for promiscuous self-association (indicated as yellow background). For clarity, the cartoon only shows three neighboring N-protein dimers, although higher-order oligomers assemble in RNPs. Nucleic acid binding sites (purple triangles) preferentially bind single-stranded RNA at the NTD (gray lines), and double-stranded RNA at the two sites per CTD dimer, with the ability to cross-link neighboring dimers potentially in various configurations. New inter-dimer interactions evolved in variants of concern are indicated by red connectors, including the promotion of beta-sheet oligomerization through the N:P13L mutation in the N-arm (as in Omicron and Lambda variants), and the introduction of cysteines at the base of the LRS helices in N:G214C (as in Lambda variants) and N:G215C (as in Delta variants). (**C**) Three-dimensional cartoon of the circular organization of N-protein domains in RNPs, with one dimer shaded slightly darker to highlight the dimeric building blocks. For clarity, subunit sizes are not drawn to scale. Alternate arrangements are depicted in **Figure 1—figure supplement 1**. (**D**) CD spectra of N-protein in the presence of SL7 under near physiological salt conditions leading to majority assembly of RNPs. Spectra are corrected for free SL7 contributions. Shown are spectra of ancestral N-protein alone (black), in the presence of SL7 forming RNPs (red), $N_\lambda$ alone (cyan), and in the presence of SL7 forming RNPs (magenta). Spectra are truncated at <205 nm due to limited buffer transparency. For comparison, the dotted line shows a previously published CD spectrum of ancestral N-protein in low-salt buffer that permits measurement at shorter wavelengths (**Nguyen et al., 2024**). Triplicate scans yield average standard deviations of 0.13 (N), 0.17 (N+SL7), 0.16 ($N_\lambda$), and 0.21 ($N_\lambda$ +SL7) $10^3$ deg cm²/dmol, respectively, with non-overlapping confidence bands for the different species, for example, between 215 and 220 nm.

The online version of this article includes the following figure supplement(s) for figure 1:

**Figure supplement 1.** Possible heterogeneity of RNPs from alternate configurations of N dimer subunits.

**Figure supplement 2.** Structural prediction of an RNP.

can modulate basic biophysical properties including its oligomeric state, thermodynamic stability of its two folded domains, LLPS propensity, charge distributions, and secondary structure content (**Nguyen et al., 2024**; **Zhao et al., 2022**), as well as its interactions with RNA (**Cubuk et al., 2024**; **Dhamotharan et al., 2024**), kinases (**Johnson et al., 2022**; **Syed et al., 2024**) and other host proteins through altered SLiMs (**Li et al., 2025**; **Ren et al., 2024**; **Schuck and Zhao, 2023**; **Tugaeva et al., 2023**).

**Table 1.** Overview of N-protein mutant species studied.

| Designation | N-protein mutations | In set of defining VOC mutations* | Predominant oligomeric state at low µM concentrations | Reaction boundary s-value in RNP assay (S) | MP final average Mw in 500–1500 kDa range (kDa) | Best-fit mass increase from 0.3 to 3 µM (kDa) | Best-fit effective RNP life-time (s) |
|---|---|---|---|---|---|---|---|
| N (ancestral) | none | Wuhan-Hu-1 | Dimer | 19.7 | 614 | 63.8 | 66.3 |
| N:P13L | P13L | $\lambda$, $o$ (all) | | | | | |
| N:Δ31–33 | Δ31–33 | $o$ (all) | | | | | |
| N:P13L/Δ31–33 | P13L, Δ31–33 | $o$ (all) | Dimer | 20.2 | 625 | 118.2 | 43.6 |
| N:R203K/G204R | R203K, G204R | α, γ, $\lambda$, $\varsigma$, and $o$ (except XEC) | Dimer | 19.5 | 596 | 70.3 | 54.6 |
| N$_o$ | P13L, Δ31–33, R203K, G204R | $o$ (except XEC) | Dimer | 20.5 | 617 | 77.2 | 58.5 |
| N:G215C | G215C (reduced) | δ (all 21 J) | Dimer/tetramer | 20.4 | 619 | 47.8 | 231 |
| N:G215C* | G215C (oxidized) | δ (all 21 J) | Tetramer | 20.8 | 649 | 56.1 | 41.8 |
| N$_\lambda$ | P13L, R203K, G204R, G214C (reduced) | $\lambda$ | Dimer/tetramer | 21.0 | 669 | 20.0 | 67.4 |
| N$_\lambda$* | P13L, R203K, G204R, G214C (oxidized) | $\lambda$ | Tetramer | 21.5 | 660 | 51.6 | 98.2 |
| N:R203M | R203M | δ, κ | | | | | |
| N$_{210-419}$* | Δ1–209 | $o$ (not XEC) | | | | | |

*Referring to the most common mutations in the variants of concern, excluding sporadic spontaneous reversions or other variations.

In the present work, we study the eponymous function of N-protein, which is the spatial condensation of the long genomic RNA (gRNA) into ribonucleoprotein particles (RNPs) for viral assembly. N-protein has two folded domains (the nucleic acid binding domain and the C-terminal dimerization domain, NTD and CTD, respectively), and three intrinsically disordered regions that include the linker between NTD and CTD, and the N-terminal and C-terminal extensions, N-arm and C-arm, respectively (*Figure 1A*; *Chang et al., 2006*). The intrinsically disordered regions amount to ≈45% of total residues, which renders N-protein highly flexible with a radius of gyration fluctuating from 5 nm to >8 nm (*Różycki and Boura, 2022*). RNPs are ≈15 nm diameter particles composed of ≈10–15 N-proteins that bind stretches of gRNA (*Klein et al., 2020*; *Yao et al., 2020*). 30–40 RNPs are distributed like beads on a string in the ≈100 nm virion (*Klein et al., 2020*; *Yao et al., 2020*). RNPs appear highly heterogeneous in electron microscopy (EM; *Carlson et al., 2022*; *Landeras-Bueno et al., 2025*; *Yao et al., 2020*), and as of now, a high-resolution structure has not been determined. The Morgan laboratory has described an in vitro model for the assembly of RNPs, where N-protein in the presence of stem-loops of the 5'-UTR RNA readily forms polymorphic ribonucleoprotein complexes that match in size, symmetry, and RNA content what would be expected for the assembled RNPs in virions (*Carlson et al., 2022*; *Carlson et al., 2020*). In conjunction with biophysical experiments and point mutations exposing essential binding interfaces, this has allowed us recently to develop a coarse-grained structural model of RNPs (*Zhao et al., 2024*). To examine how architecture and energetics of RNP assemblies can be impacted by N-protein mutations, we study a panel of N-proteins derived from ancestral Wuhan-Hu-1 and different variants of concern, including Alpha, Delta, Lambda, and Omicron (see *Table 1*), in biophysical experiments, VLP assays, and mutant virus. Specifically, we ask how the RNP size distribution and life time is modulated by: (1) the novel binding interface created by the P13L mutation of Omicron; (2) enhancements of other weak self-association interfaces through G215C of Delta and G214C of Lambda; (3) the ubiquitous R203K/G204R double mutation of Alpha, Lambda, and Omicron. We also test whether the P13L mutation improves viral fitness, similar to G215C and

R203K/G204R. The results are discussed in the framework of fuzzy complexes and molecular evolution of N in the course of viral adaptation to the human host. Understanding the salient features of the binding interfaces in viral assembly and their evolution expands our foundation for the design of therapeutics such as assembly inhibitors.

## Results

### Essential protein-protein interfaces in RNP assembly

Based on biophysical experiments of protein-protein and protein-NA interfaces of full-length (FL) N-protein and domain constructs, in combination with the in vitro RNP assembly assay (*Carlson et al., 2022*; *Carlson et al., 2020*) we have recently proposed a coarse-grained structural model of RNPs that satisfies all known binding interfaces. These include at least two known sites of protein-protein and two for protein-nucleic acid interactions distributed throughout most of the protein (*Zhao et al., 2024*; *Figure 1B and C*). N-proteins are constitutive dimers with $K_D$ in the low nM range by virtue of domain-swapped beta hairpins in the CTD (*Chang et al., 2006*; *Cubuk et al., 2025*; *Yu et al., 2005*; *Zhao et al., 2021*; *Zinzula et al., 2021*). N dimers exhibit only ultra-weak further self-association with ≈1 mM $K_D$ (*Yu et al., 2005*; *Zhao et al., 2022*; *Zhao et al., 2021*), although nucleic acid impurities can artificially scaffold N-protein into higher oligomers (*Carlson et al., 2020*; *Tarczewska et al., 2021*). However, higher-order assembly is initiated by the occupation of the nucleic acid binding site in the NTD, which allosterically promotes a helical conformation of the leucine-rich sequence (LRS) in the intrinsically disordered linker, allowing LRS helices to form promiscuous coiled-coil oligomers stabilized through hydrophobic interactions with low µM $K_D$ (*Zhao et al., 2023*). As demonstrated with the LRS point mutant N:L222P that abrogates these transient helices, LRS oligomerization forms the basis for oligomerization of N-protein dimers into RNPs, for example, dodecamers as hexamers of dimers. These oligomers are stabilized by multi-site interactions with double-stranded RNA simultaneously in two sites of the CTD dimer, and by binding preferentially to single-stranded RNA to each of the NTDs (*Cubuk et al., 2024*; *Iserman et al., 2020*; *Korn et al., 2023*; *Roden et al., 2022*), with both RNA interactions allowing inter-dimer interactions that crosslink N-protein dimers within the RNP (*Figure 1C*; *Zhao et al., 2024*). Additional stabilizing protein-protein interfaces may exist in the C-arm (*Carlson et al., 2022*; *Ye et al., 2020b*), although we have been unable to detect self-association of the C-arm by analytical ultracentrifugation up to low mM concentrations (*Zhao et al., 2023*).

As suggested in the cartoon of *Figure 1C*, this supports the hypothesis of a three-dimensional arrangement with a central LRS oligomer with symmetry properties and dimensions similar to low resolution EM images of model RNPs (*Carlson et al., 2022*; *Carlson et al., 2020*) and cryo-ET of RNPs in virions (*Klein et al., 2020*; *Yao et al., 2020*). It should be noted, however, that the arrangement sketched in *Figure 1C* is not unique and other subunit orientations could be envisioned that satisfy all constraints from experimentally observed binding interfaces, including different oligomers and anti-parallel subunits as illustrated in *Figure 1—figure supplement 1*. Extending previous ColabFold structural predictions that show multiple N-protein dimers self-assembled via the LRS coiled-coils (*Zhao et al., 2023*), we attempted the AlphaFold modeling of RNPs combining multiple N dimers with SL7 RNA ligands, mimicking our biophysical assembly model. Current AlphaFold restrictions limit the prediction to pentamers of N-protein dimers with 10 copies of SL7 RNA. While only inconsistent results were obtained – which is not surprising given the large intrinsically disordered regions exceed the predictive power of AlphaFold – some models did produce an overall RNP organization similar to *Figure 1C*, suggesting such an arrangement is at least sterically reasonable with regard to possible N-protein subunit orientations in an RNP (*Figure 1—figure supplement 2*).

Using the in vitro RNP assembly model of N-protein in mixtures with stem-loop RNA SL7, we measured secondary structure content in circular dichroism (CD) experiments (*Figure 1D*). While a significant structural change in the RNA when bound to N-protein can be deduced in the near UV, after correction for RNA contributions, in the far-UV the CD spectrum of the RNP mixtures is very similar to that of N-protein alone. A small gain in helicity can be discerned for RNP assembly mixtures at ≈220 nm, consistent with the expected coil transition in the LRS stabilizing the RNP. Similarly, small increases in helicity can be discerned for LRS helix-stabilizing cysteine mutants. It is noteworthy that the CD spectrum of the RNP mixture appears to be dominated by disordered chains. Although their signature minimum ellipticity at ≈200 nm cannot be observed directly in the high-salt buffer required

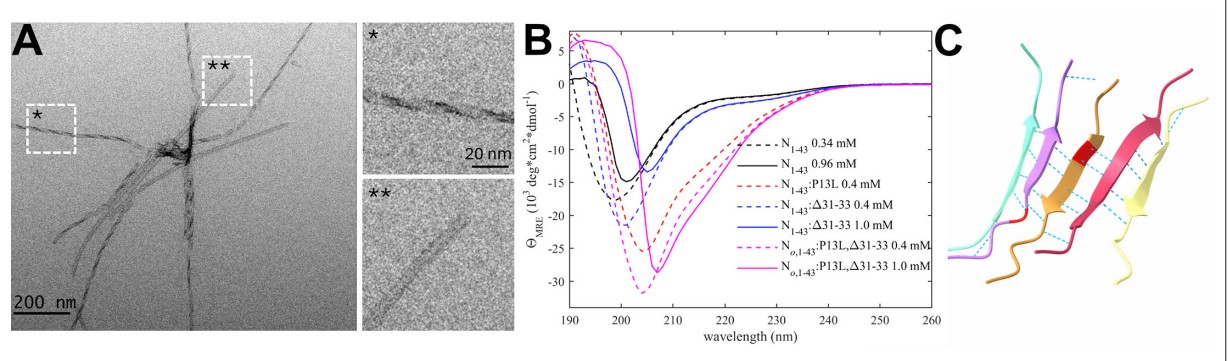

**Figure 2.** The P13L mutation creates self-association interfaces in the N-arm through stabilization of b-sheets. (**A**) Electron micrograph of negatively stained omicron N-arm $N_{o,1-43}$:P13L,Δ31-33 after equilibration at 10 μM in 20 mM HEPES, 150 mM NaCl, pH 7.50. The magnified regions are examples of twisted (*) and straight (**) fibrils. (**B**) CD spectra of $N_{1-43}$:P13L (red), $N_{1-43}$:Δ31-33 (blue), and Omicron $N_{o,1-43}$:P13L,Δ31-33 (magenta) at 0.4 mM (dashed) and 1.0 mM (solid), in comparison with ancestral N-arm (black). (**C**) Subset of ColabFold prediction of multimers of $N_{10-20}$:P13L highlighting hydrogen bonds. The P13L residue is highlighted in red in the middle peptide.

The online version of this article includes the following figure supplement(s) for figure 2:

**Figure supplement 1.** Electron micrographs of N1-43:P13L and control.

**Figure supplement 2.** Comparison of WT and P13L N-arm structure predictions.

for RNP assembly, the data are quantitatively consistent in their strongly decreasing slope from 205 to 210 nm with the previously measured disordered signature of N-protein in low-salt buffer (without RNP formation). This result is consistent with the absence of significant structure formation in the RNP, and with N-protein intrinsically disordered regions retaining most of their disorder outside the LRS.

As is depicted in the diagram *Figure 1B*, assembly of approximately 6 N-protein dimers and ≈500 bp of RNA requires simultaneous binding at multiple protein-protein and protein-RNA interfaces. Many of these interactions are weak, but they are multivalent and act cooperatively (*Zhao et al., 2024*). This feature may aid in dynamic assembly and disassembly and shape the ensemble of complex states to efficiently populate functioning RNPs. As we describe in the following, mutations of N-protein have led to diverse mechanisms modulating and promoting RNP formation through their effect on protein-protein interfaces relevant to RNP stability.

## A novel self-association interface through transient β-sheets enabled by the P13L mutation

In early epidemiological analyses, the N:P13L mutation has been identified as the most important driver for N-protein fitness, and it has become an obligatory mutation of all Omicron variants (*Obermeyer et al., 2022*; *Oulas et al., 2021*). In our recent survey of biophysical effects of N-protein mutations relative to the ancestral protein, we observed a distinct ability of the N-arm mutant peptide $N_{1-43}$:P13L to form large assemblies at ≈mM concentrations (which is not seen with the ancestral peptide), while full-length N:P13L exhibited enhancement of LLPS (*Nguyen et al., 2024*), both indicative of weak interactions. Furthermore, after prolonged storage of P13L N-arm peptide solutions at ≈mM concentrations at 4 °C, an increase in the solution viscosity was observed (*Nguyen et al., 2024*). In the present work, we studied these effects of the P13L mutation in more detail. As shown in *Figure 2A*, negative-stained EM images show the formation of fibrils of N-arm peptides of Omicron variant $N_{1-43}$:P13L,Δ31–33. Similarly, fibril formation was observed for $N_{1-43}$:P13L N-arm peptide lacking the deletion Δ31–33, but not in controls with ancestral N-arm, the ancestral N-arm carrying only the deletion $N_{1-43}$:Δ31–33, or the disordered ancestral C-arm (*Figure 2—figure supplement 1*). Thus, the N-arm mutation P13L is responsible for the formation of fibrils in N-arm peptides after prolonged storage. Some of these N-arm fibrils exhibit a twisted morphology with a width of ≈5 nm (*Figure 2A*), in some instances exhibiting patterns of strand breaks. Such fibrils are frequently encountered in proteins that can stack β-sheets, such as in amyloids (*Paravastu et al., 2008*). While we have not observed fibril formation in the context of full-length N, and have no evidence such fibrils are physiologically relevant,

their occurrence in solutions of truncated N-arm peptide nonetheless demonstrates the introduction of ordered N-arm self-association interfaces in conformations of P13L mutants.

Based on NMR studies of the N-arm, Zachrdla et al. previously reported a propensity for residues 13–19 to transiently populate extended/β-structure-like conformations (*Zachrdla et al., 2022*), which we hypothesized may be strengthened through the P13L mutation. To test the formation of β-sheet structure in N:P13L, we carried out CD experiments of ancestral N-arm $N_{1-43}$, and mutants carrying the P13L mutation, or Δ31–33, or both in the Omicron variant $N_{1-43}$:P13L,Δ31–33 (*Figure 2B*). While the ancestral N-arm at ≈1 mM (≈4.6 mg/ml) concentrations exhibits CD spectra with a minimum at ≈200 nm typical of disordered conformations (black), the Omicron N-arm has a significantly higher structure content (magenta), consistent with β-sheets, as revealed by the strong negative mean residue ellipticity in the 210–220 nm range. Interestingly, diluting the 1 mM sample (solid) to a concentration of 0.4 mM (dashed) reveals a large shift in the far-UV spectra from positive to negative ellipticity at ≈200 nm, as well as a shift in the minimum to lower wavelengths, both indicative of a significant increase of disorder upon dilution. This is consistent with the stabilization of β-sheets in a reversible, strongly cooperative self-association process with an effective $K_D$ in the high μM to low mM range. Dissecting the origin of the increased β-sheet content, it is apparent that the majority of the effect arises from the P13L mutation (red) alone, with a minor contribution by Δ31–33.

Finally, confirming the interpretation of the EM images and the CD data, as well as the β-structure propensity reported from NMR data (*Zachrdla et al., 2022*), the structural prediction of $N_{10-20}$:P13L in ColabFold displayed oligomers with stacking β-sheets of residues 12–18 with typical hydrogen bond patterns (*Figure 2C*), whereas ancestral peptides did not lead to well-organized structures (*Figure 2—figure supplement 2*). Analogous predictions of N-arms with two other frequent mutations, P13S and P13T, did not lead to β-sheet structures as in P13L.

While this self-association interface in the P13L N-arm is weak and its direct observation in biophysical experiments requires mM concentrations, which far exceed average intracellular concentration of N, such weak interactions can become highly relevant physiologically when high local concentrations are prevailing, for example, when the disordered extension is preconcentrated while tethered within macromolecular assemblies as in the RNP, or in macromolecular condensates.

## Enhanced oligomerization of the leucine-rich sequence through cysteine mutations

The protein-protein interaction interfaces driving LRS self-association have been studied in great detail in biophysical experiments and MD simulations for the ancestral molecule and several mutations in the mutational landscape (*Zhao et al., 2022*). A key feature is a pattern of hydrophobic residues that can combine to form a transient helix creating a hydrophobic surface stretching from ≈222–234 on one side of the helix. We have previously discovered the self-association of the LRS after analysis of the mutational landscape (*Zhao et al., 2022*) and reports of a conspicuous defining mutation N:G215C in Delta variants that correlated with its rise in 2021 among clades with identical spike mutations (*Marchitelli et al., 2021*; *Stern et al., 2021*; *Zhao et al., 2022*). As deduced from MD simulations, the cysteine at position 215 is located at the base (N-terminal end) of the transient helix and, through its lower flexibility than the glycine, serves to redirect the adjacent upstream disordered residues such that helices are more prone to form stabilizing coiled-coil interactions (*Zhao et al., 2023*; *Zhao et al., 2022*). As shown experimentally by sedimentation velocity analytical ultracentrifugation (SV-AUC) in reducing conditions, this enhances non-covalent self-association of LRS peptides as well as full-length N dimers by 2–3 orders of magnitude (*Zhao et al., 2023*; *Zhao et al., 2022*). Covalent disulfide bonds in the LRS in non-reducing conditions were found to further promote LRS oligomerization. However, there is no conclusive data yet whether covalent bonds in the LRS occur in vivo, or any G215C effect is entirely non-covalent due to the significant strengthening of LRS helix oligomerization (see Discussion). In any event, the G215C mutation leads to enhanced assembly in a VLP assay (*Zhao et al., 2024*), and as shown in reverse genetics experiments, in vivo confers a significant replication advantage and an altered virion morphology (*Kubinski et al., 2024*).

Here, we studied a mutation of G214, which in the mutational landscape exhibits a similar mutation pattern as G215. In particular, we focus on an independent introduction of a cysteine in the LRS that occurred in the Lambda variant, prevalent in South America in 2020–2021 (*Wink et al., 2022*), with the defining N-protein mutations P13L, R203K/G204R, and G214C (*Table 1*). (We will adopt a nomenclature

where the complete set of defining mutations of a variant will be referred to by its Greek letter, that is N:P13L/R203K/G204R/G214C is $N_\lambda$, and analogously the set of Omicron mutations N:P13L/Δ31–33/R203K/G204R are referred to as $N_o$; see **Table 1**). The effect of the G214C mutation is unknown. Due to the close proximity of 214 and 215, we asked whether it enhances LRS self-association similarly to G215C, and to this end first synthesized a LRS peptide comprising $N_{210-246}$:G214C. Unexpectedly, unlike the chemically identical $N_{210-246}$:G215C peptide, it exhibited low solubility. This prohibited its characterization at sufficiently high concentrations for study of self-association by SV-AUC, which required in excess of ≈0.4 mM for $N_{210-246}$:G215C peptides to display oligomers. However, dynamic light scattering (DLS) revealed the presence of $N_{210-246}$:G214C complexes with hydrodynamic radii ranging from 6 to 40 nm (in comparison to 1–2 nm for $N_{210-246}$:G215C **Zhao et al., 2022**) in reducing conditions, and slightly larger in non-reducing conditions (**Figure 3—figure supplement 1**). For $N_{210-246}$:G214C, a cumulant analysis results in radii of 8.8 nm and 10.6 nm and polydispersity indices of 0.40 and 0.35 for reducing and non-reducing conditions, respectively. This shows that while G214C also strengthens the LRS self-association interface, it exhibits different properties compared to G215C.

To gain more insight into the different behavior of the cysteine LRS mutants, we carried out MD simulations of monomeric and trimeric LRS peptides $N_{210-246}$. As shown in **Figure 3**, both cysteine mutants extend the helix by stabilizing the flexible GG motif into a well-defined a-helix turn. Importantly, however, the sulfhydryl groups in position 214 vs 215 assume different orientations relative to the hydrophobic patch serving as the oligomerization interface. Under reducing conditions, this has the potential to also cause variation in physicochemical properties of resulting oligomers, for example through repositioning of the adjacent glutamate E216 and resulting changes in the surface electrostatic potential, and to thereby alter the oligomerization scheme (**Figure 3—figure supplement 2**). It can be expected that under oxidative conditions, the differences between 214 C and 215 C are exacerbated due to their different relative orientation of the hydrophobic interface of the helix and the sulfhydryl groups, leading to different oligomeric states and different phase separation properties.

## Impact of enhanced self-association interfaces on the oligomerization of full-length N

In principle, since N-protein is a constitutive dimer coupled in the CTD, binding interfaces in the LRS and N-arm may form intra-dimer bridges simply further stabilizing the dimeric state. Indeed, coarse-grained molecular simulations of N-protein dimers have revealed large conformational fluctuations where the LRS, for example, can make rare intra-dimer contacts (**Różycki and Boura, 2022**). On the other hand, given the large flexibility of the disordered chain, the same interfaces may lead to higher oligomers if they establish one or two inter-dimer bridges across different dimers, generating different classes of tetramers. We would expect a concentration-dependent probability of such inter-dimer contacts for N-protein in solution.

Experimentally, in the absence of nucleic acid ligands, ancestral N-protein at low µM concentrations is essentially dimeric with only hints of reversible higher-order oligomers (**Yu et al., 2005**; **Zhao et al., 2021**). With the strengthened LRS helix stability through the G215C mutation, in reducing conditions, we observe reversible tetramerization with a $K_D$ of 1.0 (0.7–1.5) µM (**Figure 4A**), consistent with previous work (**Zhao et al., 2022**). We made analogous observations by SV-AUC for $N_\lambda$ (containing the G214C mutation) in reducing conditions, demonstrating it similarly can form reversible inter-dimer bonds (**Figure 4A**). Clearly, strengthened LRS helices can alter the N-protein oligomeric state by crosslinking dimers into tetramers.

To study the potential impact of disulfide bonds on N-protein oligomeric state, we prepared oxidized full-length N:G215C and $N_\lambda$ by extensively dialyzing the reduced protein in TCEP-free buffer while purging air through the dialysate for gentle passive oxygenation. This oxidized sample is referred to as N:G215C* and $N_\lambda$*. Using a DTNB assay, we assessed the percentage of free sulfhydryl groups to be ≈30% for both N:G215C* and $N_\lambda$*, respectively. Since the LRS mutation provides the sole cysteine in N-protein, we concluded that the majority of N:G215C* and $N_\lambda$* is disulfide-linked in the LRS, which was confirmed by non-reducing SDS-PAGE (**Figure 4—figure supplement 1**). Due to the non-covalent dimerization in the CTD as well as non-covalent tetramerization via LRS interfaces (see above), these samples may be complex mixtures of dimers and/or higher oligomers with different patterns of covalent and non-covalent intra- and inter-dimer LRS interactions. Using SV-AUC, we established that close to half of both the N:G215C* and $N_\lambda$* sample is tetrameric at low µM concentrations

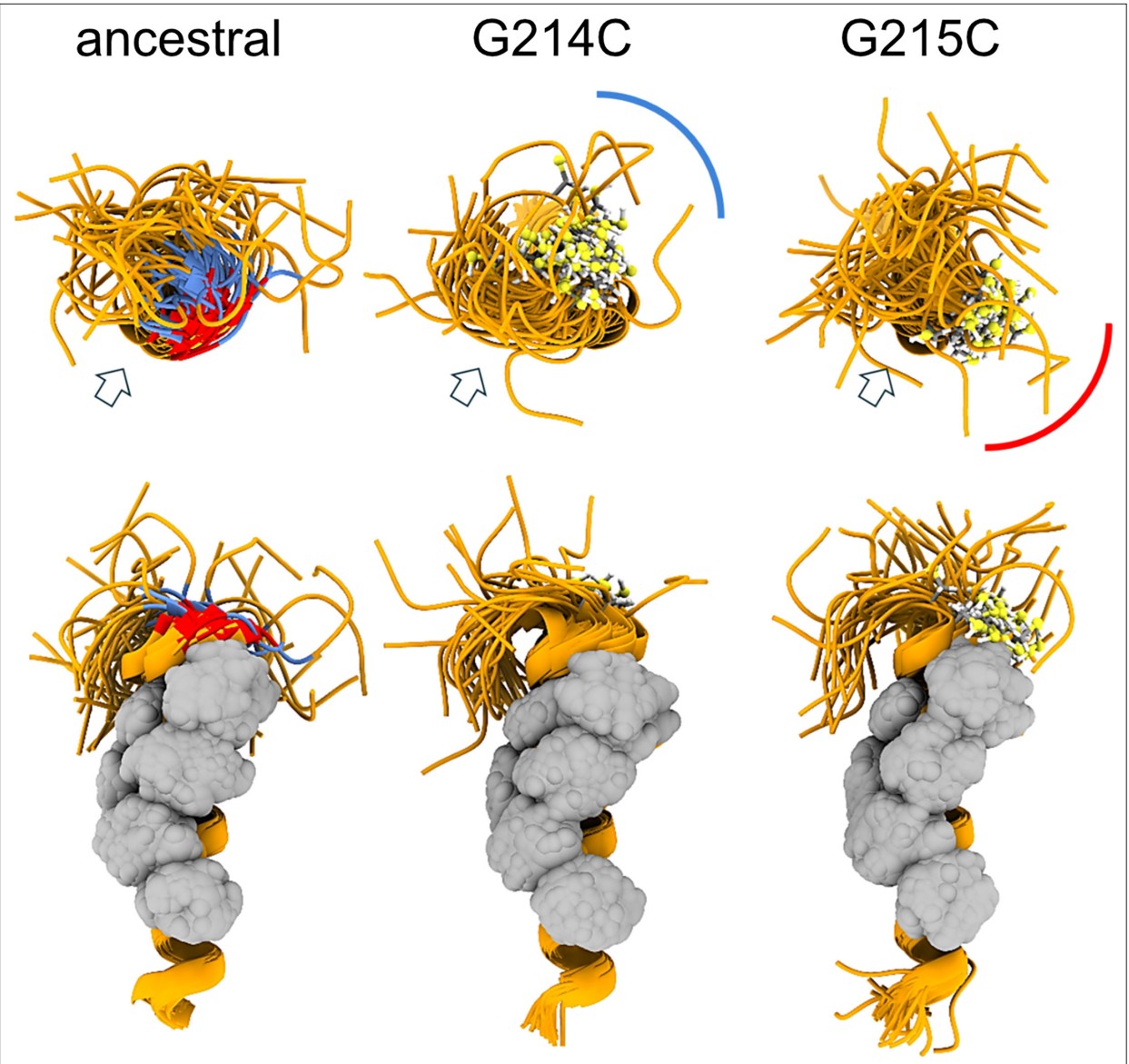

**Figure 3.** MD simulation of ancestral LRS and comparison with 214 C and 215 C mutants of Delta and Lambda variants. Snapshots at equal time intervals (4 ns) taken from the 200-ns MD simulations of the ancestral peptide N$_{210-246}$ and single-point G→C mutants (all monomers were oriented by overlying the helical region and displayed in orthosteric view). The upper row shows a view from the N-terminus side (with the helix axes perpendicular to the plane of the figure), while the lower row presents a side view (with axes on the plane). For clarity, the disordered C-terminal segment (residues 236–246) has been removed. The glycine and cysteine residues at positions 214 and 215 (colored blue and red, respectively, in the ancestral peptide and rendered as ball-and-stick in the mutants) restructure the flexible backbone around the GG motif into a well-defined α-helix turn, directing the sulfhydryl group in specific orientations (illustrated schematically by the blue and red arches). These orientations can be quantified relative to the Leu-rich central region (indicated by arrows and rendered as gray van der Waals spheres), which forms the hydrophobic interfaces of the oligomers (*Zhao et al., 2023*). Under reducing conditions, this reorientation of the N-terminus relative to the helix can influence helix binding during the early stages of oligomerization or alter the conformation and physicochemical properties of the resulting oligomers, as illustrated in *Figure 3—figure supplement 2*.

The online version of this article includes the following figure supplement(s) for figure 3:

**Figure supplement 1.** Size distribution of G214C cysteine mutant LRS peptides.

**Figure supplement 2.** Structural and physicochemical effects of LRS mutants G214C and G215C under reducing conditions.

(*Figure 4A*). Because oxidation significantly increases the tetramer population, we conclude that one or two covalent inter-dimer bonds can form that enhance N-protein oligomerization.

In contrast to the modulation of the coiled-coil LRS interfaces, the de novo creation of the N-arm self-association interface through beta-sheet interactions enabled by P13L cannot be readily observed

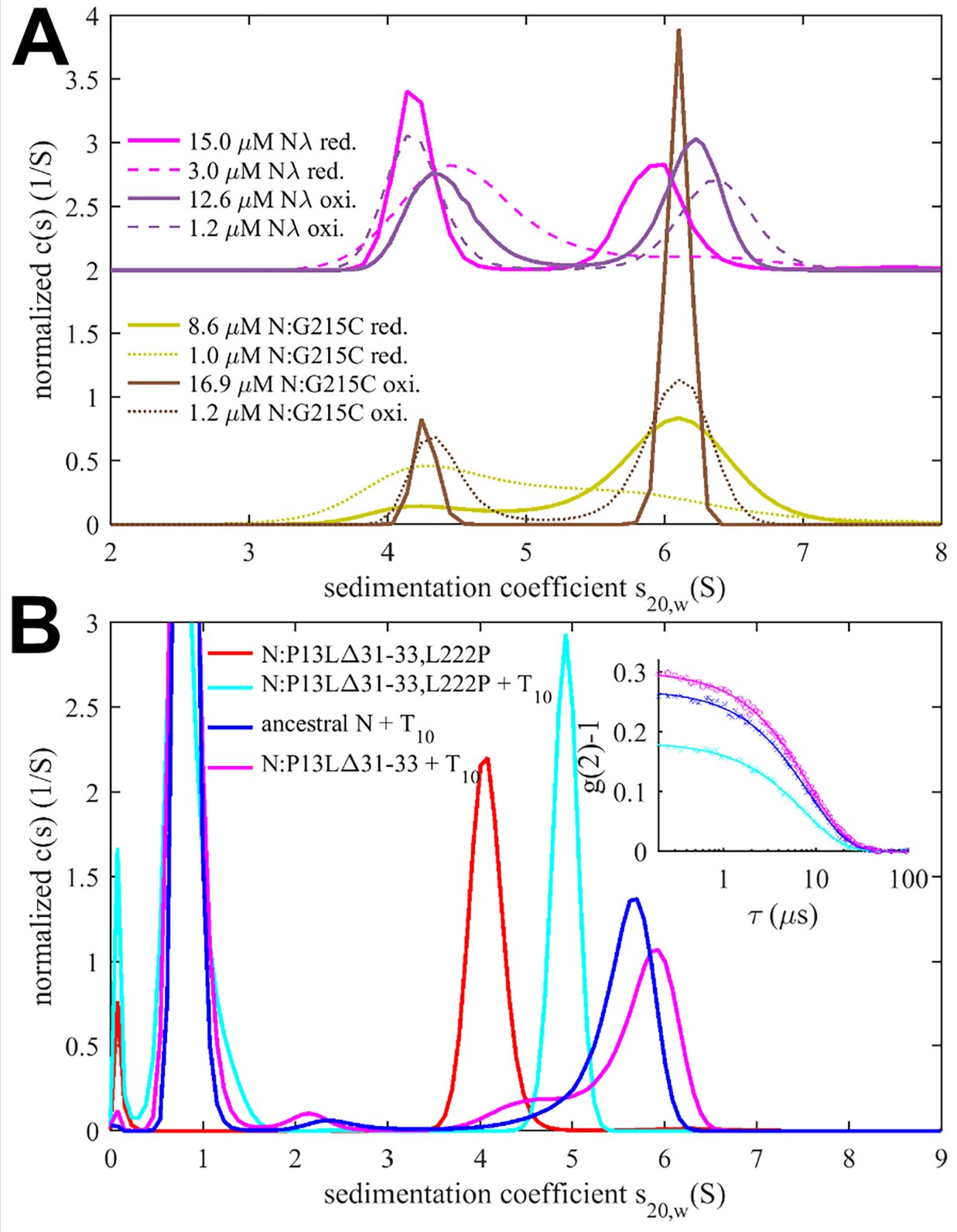

**Figure 4.** Impact of mutations in self-association interfaces on oligomeric state. (**A**) Sedimentation coefficient distributions of cysteine mutants N:G215C (reduced in yellow) and N:G215C* (oxidized in brown), as well as reduced $N_\lambda$ (reduced in magenta) and $N_\lambda$* (oxidized in violet), with $N_\lambda$ data offset by 2. For each sample, data were acquired at high (solid lines) and low (dashed lines) concentration. (**B**) Sedimentation coefficient distributions of 2 μM

*Figure 4 continued on next page*

*Figure 4 continued*

N:P13L,Δ31-33, ancestral N, and N:P13L,Δ31-33,L222P in the presence of 10 μM T$_{10}$ in low-salt buffer. The inset shows DLS autocorrelation data of the same samples (symbols) and single-species fits (lines).

The online version of this article includes the following source data and figure supplement(s) for figure 4:

**Figure supplement 1.** Non-reducing SDS-PAGE of reduced and oxidized N:G215C* and N$_λ$*.

**Figure supplement 1—source data 1.** Labeled non-reducing SDS-PAGE of reduced and oxidized N:G215C* and N λ *.

**Figure supplement 1—source data 2.** Unlabeled non-reducing SDS-PAGE of reduced and oxidized N:G215C* and N λ *.

**Figure supplement 2.** N-arm mutations rescue LLPS of LRS-helix deficient mutants.

**Figure supplement 3.** Measurement of the affinity of N:P13L,Δ31-33 for oligonucleotide T$_{10}$ by SV-AUC titration.

in full-length N-protein at low μM concentrations. Similar to the ancestral LRS interface, it provides only ultra-weak binding energies that require mM concentrations to significantly populate oligomers. This is fully consistent with the previous observation by SV-AUC that neither N:P13L,Δ31-33 nor N$_o$ with the full set of Omicron mutations show any significant higher-order self-association at low μM concentrations, whereas at high local concentrations – as observed in phase-separated droplets – they can modulate and cooperatively enhance self-association processes (*Nguyen et al., 2024*). (In fact, P13L can substitute for the LRS promoting LLPS, as observed in the rescue of LLPS by N:P13L,Δ31-33/L222P mutants, whereas N:L222P LRS-abrogating mutants are deficient in LLPS; *Figure 4—figure supplement 2*) Another process that increases the local concentration of N-arm chains is the tetramerization of full-length N-protein. As described earlier, occupancy of the NA-binding site in the NTD allosterically promotes self-assembly of the LRS into higher oligomers (*Zhao et al., 2021*). We hypothesized that these oligomers may be cooperatively stabilized by additional N-arm interactions in P13L mutants.

To this end, we carried out SV-AUC and DLS experiments of 2 μM ancestral N and the full-length N-arm mutant N:P13L,Δ31-33 in the presence of a short oligonucleotide T$_{10}$ that occupies the NTD binding site but is too short for scaffolding multiple N-protein dimers (*Zhao et al., 2021*; *Figure 4B*). Low-salt buffer (10 mM NaCl, 2.7 mM KCl, 10.1 mM Na$_2$PO$_4$, 1.8 mM KH$_2$PO$_4$, pH 7.4) in this experiment ensures maximal occupancy of the NTD binding site for nucleic acid in solutions with 10 μM T$_{10}$, which under these conditions has a K$_D$ below 0.1 μM (*Zhao et al., 2021*). In a control experiment by SV-AUC, we measured the affinity of the T$_{10}$ oligonucleotide for ancestral N and N:P13L,Δ31-33 and observed no significant difference in moderate ionic strength (*Figure 4—figure supplement 3*). Ancestral N protein in the absence of oligonucleotide sediments at ≈4.0 S, reflecting its dimeric state afforded by the CTD dimerization domain (*Zhao et al., 2021*). When LRS oligomerization is abrogated through introduction of a L222P mutation, binding of T$_{10}$ to the N-protein dimer still induces a conformational change and increases its s-value to ≈4.9 S (*Zhao et al., 2023*). With the native LRS in the ancestral N, this T$_{10}$-ligated state allows LRS oligomerization, which can be observed through the formation of a reaction boundary in SV-AUC with an s-value of ≈5.6 S, reflective of a mixture of dimers and tetramers in rapid exchange relative to the time scale of sedimentation (*Schuck, 2010*; *Zhao et al., 2023*). As a control, we reproduced this previously reported result (*Figure 4B*, blue). Introduction of the N-arm mutations N:P13L,Δ31-33 causes a further increase of the s-value of the reaction boundary to ≈5.8 S, indicating an increase in the tetramer stability (*Figure 4B*, magenta). This stabilization of the tetramer is corroborated independently by an increase in the average hydrodynamic radius of N:P13L,Δ31-33 in mixture with T$_{10}$ (5.51 nm) relative to ancestral N (5.37 nm) or the LRS mutant N:P13L,Δ31-33,L222P (4.84 nm; *Figure 4B* inset). Thus, the N-arm mutation clearly strengthens inter-dimer interactions, even though the added binding energy is too weak to produce detectable tetramer populations at micromolar concentrations by itself. In principle, allosteric interactions between the distant disordered N-arm and the LRS in the disordered linker might exist that cause enhanced tetramerization of N:P13L,Δ31-33 with occupied NA-site in the NTD. However, a more parsimonious explanation is that the additional self-association interface in the N-arm created by P13L makes inter-dimer contacts that add to the separate oligomerization of the LRS helices in stabilizing tetramers.

## Mutation effects on RNP assembly and stability

As the assembly of RNPs requires the concerted effect of several binding interfaces, we asked whether the enhanced LRS coiled-coil stability and the novel N-arm self-association interface impact the RNP

stability. To examine this experimentally, we carried out in vitro RNP assembly experiments using the assay developed previously by the Morgan laboratory (*Carlson et al., 2022*; *Carlson et al., 2020*). As mentioned above, it is based on the observation that mixtures of N-protein with stem-loop RNA from the viral 5'-UTR at low µM concentrations readily form polymorphic ribonucleoprotein complexes that match in size, symmetry, and RNA content what would be expected for the assembled RNPs in virions (*Carlson et al., 2022*; *Carlson et al., 2020*). The use of stem-loop SL7 as RNA substrate helps to minimize structural polydispersity arising from variable secondary structure elements. We embarked on the experimental roadmap introduced previously (*Zhao et al., 2024*) consisting of SV-AUC experiments that hydrodynamically resolve RNPs in dynamic assembly equilibrium in solution as fast-sedimenting reaction boundaries (*Schuck, 2010*; *Schuck and Zhao, 2017*), in combination with complementary mass photometry (MP) experiments that can resolve populations of different protein/RNA complexes and their dissociation products at sub-µM concentrations through interferometric sizing of single-molecule surface adsorption events (*Wu and Piszczek, 2021*).

Sedimentation coefficient distributions of ancestral and different mutant full-length N at 3 µM in the presence of 3.4 µM SL7 are shown in *Figure 5A*. For reference, under these conditions, the RNPs of the ancestral N-protein form a reaction boundary with a weighted average sedimentation velocity of 19.7 S (black). Under otherwise identical conditions, significantly faster sedimentation can be discerned for N:P13L,$\Delta$31-33 (20.2 S, red). Faster sedimentation can reflect an increase in size of the complexes and/or increased populations and lifetimes of complexes in dynamic equilibrium, that is higher affinity and stability. By contrast, N:R203K/G204R exhibits a slightly lower *s*-value of 19.5 S (blue). Both R203K/G204R and P13L,$\Delta$31-33 combine in the set of defining Omicron mutations, $N_o$, which produces RNP boundaries at 20.5 S (cyan). It appears the N-arm mutations can more than compensate for the loss of RNP stability through the R203K/G204R mutation in the linker. The introduction of the LRS cysteine in N:G215C is nearly as effective, with an *s*-value of 20.4 S (orange). Finally, the largest increase in sedimentation velocity can be discerned for the combination of P13L/R203K/G204R with the LRS enhancing cysteine G214C in $N_\lambda$ with 21.0 S (magenta; see *Table 1*).

When these samples are diluted and equilibrated at tenfold lower concentration, the RNPs are largely dissociated, as may be discerned from the reduced amplitude of the rapidly sedimenting peak and significant population of species with *s*-values between 7 and 18 S (*Figure 5B*). This highlights the cooperative oligomerization of the N-dimer/2SL7 subunits observed previously (*Zhao et al., 2024*). Interestingly, the only partial dissociation of $N_o$, N:G215C, and $N_\lambda$ RNPs suggests the existence of a subpopulation with higher stability. By contrast, the sedimentation coefficient distribution of RNPs from 0.3 µM N:P13L,$\Delta$31-33 is more similar to that of the ancestral N-protein and N:R203K/G204R, suggesting the augmented RNP population caused by combining these mutations is kinetically not as stable and the contributions from N-arm interfaces are more transient.

Side by side to the SV experiments, MP measurements were carried out on the same samples to shed more light on the mass distribution of the RNP particles. For discrimination of individual surface adsorption events on the coverslip, the sample concentrations cannot exceed 0.3 µM protein with 0.34 µM SL7. As shown in *Figure 5C*, this results in a ladder of oligomers of N-dimer/2SL7 subunits ranging from ≈120 kDa (a single N-dimer/2SL7 subunit) to ≈700 kDa (a hexamer of subunits; *Carlson et al., 2022*; *Zhao et al., 2024*). While this is analogous to the solution data in *Figure 5B*, for the detailed comparison of SV and MP distributions, their different signal weights should be considered: where MP counts individual particles producing a number distribution, SV records a signal proportional to mass, and therefore is strongly skewed toward larger particles compared to MP. Also, in a trade-off between resolution and susceptibility to systematic errors, there is a potential for chemical properties to bias the surface adsorption, and for some adventitious experimental variation originating from the glass substrates. Nonetheless, we obtained results qualitatively consistent with the SV-AUC data, where N:P13L is largely dissociated like ancestral N-protein and N:R203K/G204R at 0.3 µM; and at the same time, $N_o$, N:G215C, and $N_\lambda$ RNPs retain more of the oligomeric state. Interestingly, N:G215C shows the least fully dissociated dimer and retains more of a pentameric species, which is in contrast to the other cysteine-containing mutant $N_\lambda$ (*Figure 5C*).

The ladder of oligomeric subunits resolved in MP poses the question whether the largely assembled state at 3 µM is uniform. Under these conditions, SV-AUC shows a single RNP reaction boundary, but it is relatively broad, which would be equally consistent with a reaction boundary from a single complex in rapid association/dissociation exchange on the time scale of sedimentation (<1000 s;

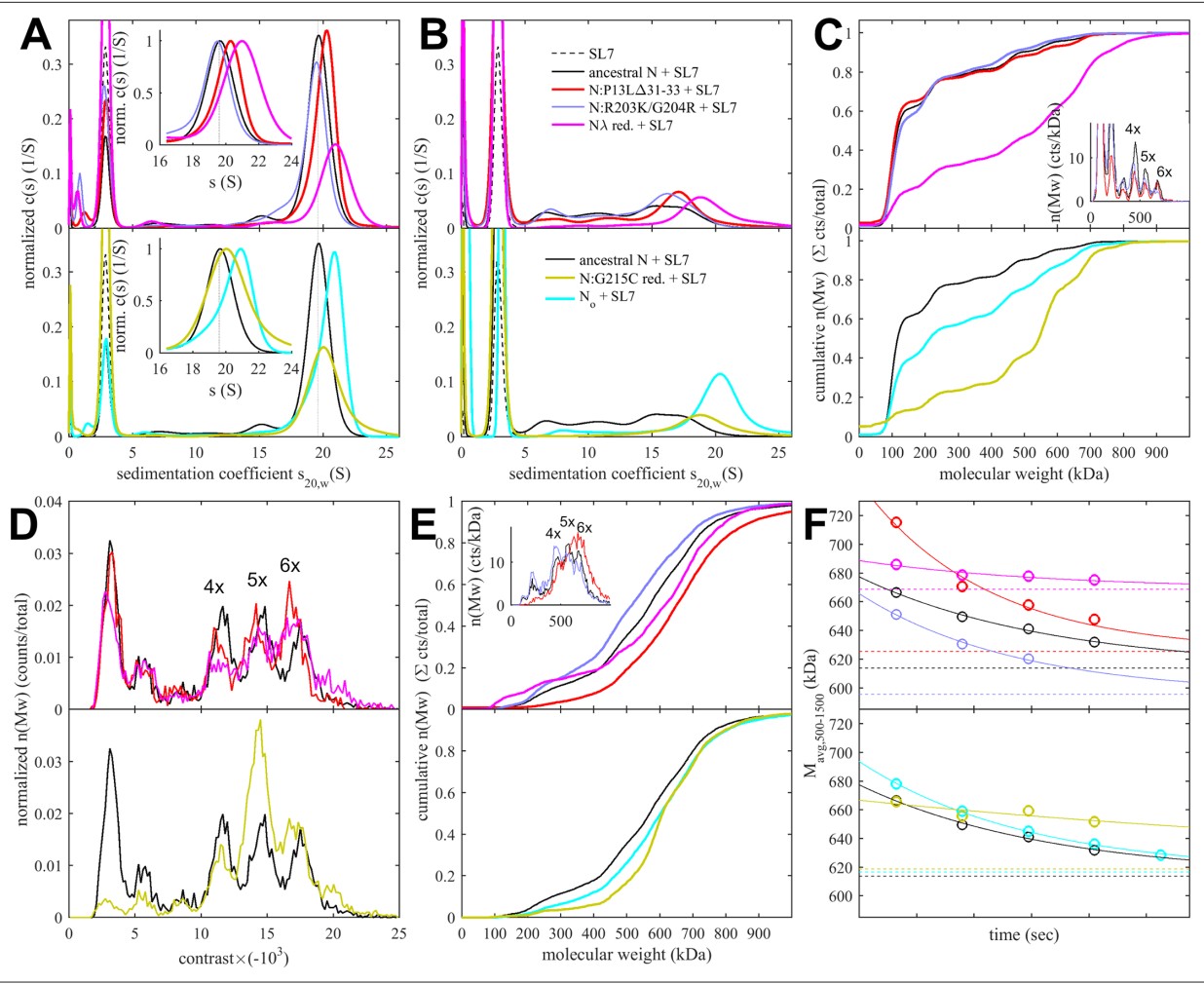

**Figure 5.** Size distributions and stability of ancestral and mutant RNPs. Show are SV (**A, B**) and MP data (**C–F**) for mixtures of N-protein with stem-loop RNA SL7 in molar ratio of 1(N):1.15(SL7) at high concentration of 3 μM (**A, D**) or low concentration of 0.3 μM (**B, C, E, F**) in reducing buffer conditions. All panels use the same color scheme for N-protein: ancestral (black), N:P13L,Δ31-33 (red), N:G215C (orange), $N_o$ (cyan), N:R203K/G204R (blue), $N_\lambda$ (magenta). All panels are subdivided into two plots for clarity, with each showing ancestral trace in black for comparison. (**A**) Sedimentation coefficient distributions of mixtures equilibrated at high concentration, with reaction boundary peaks magnified in the inset. For reference, the ancestral RNP s-value is drawn as a dotted vertical line. Absorbance data are recorded at 260 nm and are weighted by SL7 content of sedimenting species. Higher reaction boundary s-values signify greater affinity or lifetime of the mutant RNPs. (**B**) Sedimentation coefficient distributions of the same samples as in (**A**), tenfold diluted and equilibrated, highlight dissociation of most RNPs into a range of intermediate size complexes. (**C**) MP experiments of equilibrated 0.3 μM mixtures. The measured number distributions are presented as cumulative distributions, which display higher percentages of large species as shifts to the right. Most samples are largely dissociated into dimers, with remaining peaks corresponding to populations of dimers to hexamers of $N_2/SL7_2$ subunits (as highlighted in the differential distributions in the inset). As an example for the resolution of distinct species, the inset shows the differential distribution (histogram) for N:R203K/G204R (blue), ancestral N (black), and N:P13L,Δ31-33 (red), with the peak labels indicating the number of N-dimer/2SL7 subunits. (**D**) Mass distributions acquired in stopped-flow configuration applied to 3 μM mixtures. Larger (negative) contrasts correspond to higher molecular weights, with major peaks corresponding to species containing 1, 4, 5, and 6 $N_2/SL7_2$ subunits. (**E**) For kinetic experiments, mass distributions were acquired in different time intervals after tenfold dilution of 3 μM mixtures, here showing data collected from 3 s to 23 s. (**F**) Number-average molecular weights of assembled RNPs between 500 and 1500 kDa observed in consecutive 20 s data acquisition intervals after tenfold dilution of 3 μM mixtures (circles). The dashed horizontal lines are number-averages determined from the equilibrated 0.3 μM mixtures in (**E**). The solid lines are a best-fit single exponentials constrained to decay to the measured equilibrium values, yielding RNP lifetimes listed in *Table 1*.

*Schuck and Zhao, 2017*), as with a polydisperse mixture of several unresolved large oligomers. In order to gain more insight into the mass distribution under these 3 μM conditions, for some constructs, we employed a microfluidic accessory device for the MP instrument allowing rapid dilution with a dead time of <0.1 s to minimize RNP dissociation. The resulting data in *Figure 5D* again exhibit a ladder of oligomeric peaks, where the majority of N-protein assembled in a heterogeneous mixture

of RNPs between tetramer and hexamer of subunits. Thus, the dissociation products after dilution in *Figure 5C* do not seem to originate from a single assembled complex. Furthermore, we observe characteristic differences in the oligomeric distribution between different constructs. Similar to the equilibrated lower concentration conditions, relative to ancestral N, the hexamer population is augmented for N:P13L,Δ31-33, N:G215C, and N$_\lambda$. Notably, there is again a prominent pentamer population for N:G215C.

Since the disassembly of RNPs after viral entry is another critical step in the viral life cycle, we aimed to probe the kinetic stability of the RNP complexes. To this end, we applied a modified pipettor-based sample application protocol where rapid dilution of the 3 μM mixture was followed by several consecutive 20 s periods of data acquisition. The data from the acquisition immediately following the dilution (acquired between 3 and 23 s) is shown in *Figure 5E*. It provides mass distributions showing substantial assembly into heterogeneous populations of RNPs. Qualitatively consistent with the 3 μM mixtures in SV-AUC, the N:R203K/G204R has a clear destabilizing effect on the RNP, whereas all mutants with enhanced binding interfaces exhibit higher populations of larger RNPs, with the greatest enhancement for N:P13L,Δ31-33. In order to focus on the kinetics of RNP dissociation, we calculated the number average molecular weight of RNPs, which is plotted as a function of decay time in *Figure 5F* and empirically fitted as a single exponential decay attaining the separately measured equilibrium value measured at 0.3 μM (for distributions see *Figure 5C*, for number averages *Table 1*). While N:P13L,Δ31-33 produces the largest increase in RNP molecular weight between 0.3 μM and 3 μM, these RNPs have the shortest lifetime (t=44 s *vs* 66 s for the ancestral N), suggesting the added H-bond interactions in the N-arm to be rapidly reversible. On the other hand, RNPs of N:G215C have a lower average mass (being dominated by a pentameric oligomer of subunits), but show an increased kinetic stability (t=231 s). RNPs of N$_\lambda$ carrying both an LRS cysteine and the N-arm mutation are simultaneously of higher average molecular weight and persist most upon dilution (with a best-fit RNP equilibrium level of 669 kDa; *Figure 5F*).

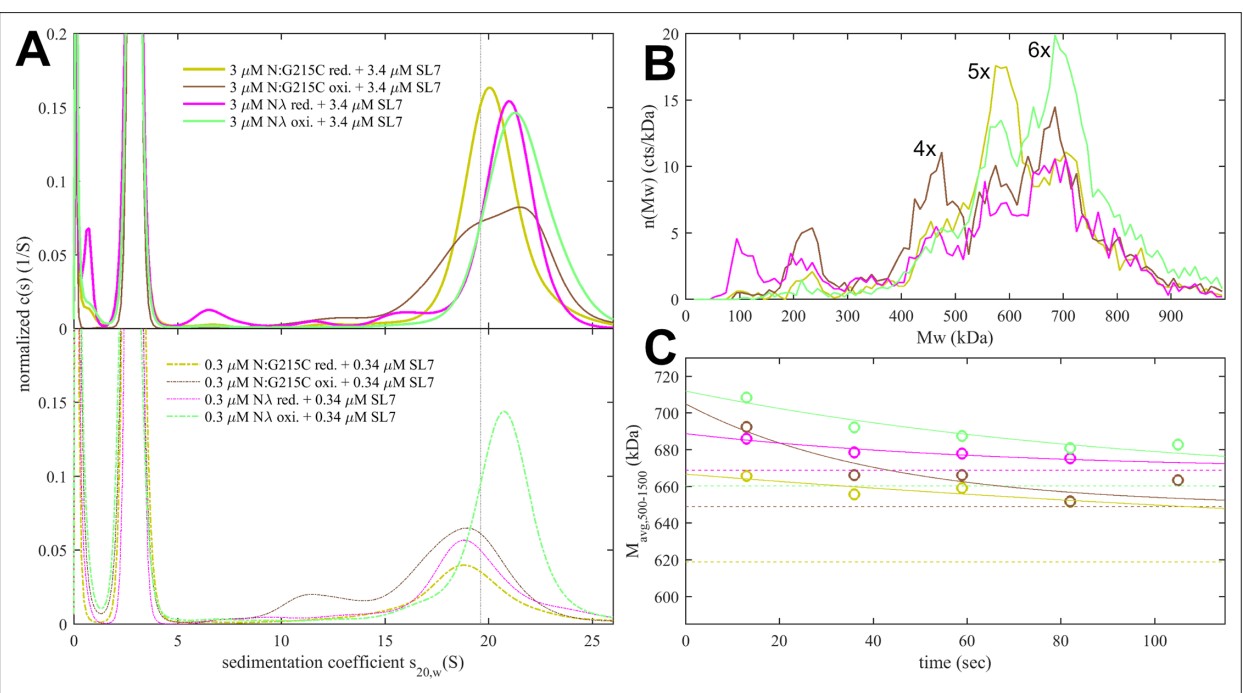

**Figure 6.** Impact of LRS disulfide bonds on the size and stability of RNPs. N-protein with cysteine mutations in the LRS was oxidized to form disulfide-linked oligomers (as shown in *Figure 4A*) and mixed with stem-loop RNA SL7 in molar ratio of 1(N):1.15(SL7). Shown are data for oxidized N:G215C* (brown) and oxidized N$_\lambda$* (green), and for comparison, reduced N:G215C (yellow) and reduced N$_\lambda$ (magenta). (**A**) Sedimentation coefficient distributions at 3 μM (upper panel) and 0.3 μM (lower panel) protein. (**B**) Molecular weight distributions in MP experiments of the same mixtures rapidly diluted to 0.3 μM protein, acquired from 3 to 23 s after dilution, with peak labels reflecting the multiples of N dimer/2SL7 subunits. (**C**) Time course of number-average RNP molecular weights between 500 and 1500 kDa (circles), determined from rapid dilution experiments in (**B**) for consecutive 20 s data acquisition intervals. The solid lines are best-fit single-exponential decays constrained to attain the separately measured equilibrium values at 0.3 μM protein (dashed lines), with lifetimes listed in *Table 1*.

Even though it is still unclear whether disulfide bonds of N cysteine mutants form in vivo, we were curious about the impact of disulfide-linked oligomers of the cysteine mutants on their RNP structure and stability in our biophysical assembly model and carried out analogous experiments with the substantially oxidized protein preparations depicted above in *Figure 4A*. As shown in *Figure 6A*, sedimentation coefficient distributions of 3 µM oxidized N:G215C* and oxidized $N_\lambda$* (carrying 214 C) in assembly mixtures with SL7, show faster reaction boundaries when oxidized as compared to their reduced form, with 20.8 S vs 20.4 S for N:G215C, and 21.5 S vs 21.0 S for $N_\lambda$. Upon dilution, the RNPs of oxidized N:G215C* protein dissociate similarly to those of reduced N:G215C. For oxidized $N_\lambda$ RNPs dissociation is less, but still significant. The molecular weight distributions in MP measured in the first 20 s after rapid dilution of the 3 µM stock (*Figure 6B*) show largely assembled mixtures of tetramers, pentamers, and hexamers of the N-dimer/2SL7 subunits, with a large increase in the population of hexameric RNPs in oxidized vs reduced forms. The tail of even higher molecular weight species is enhanced, pointing to subpopulations of heptameric species, especially for oxidized $N_\lambda$ RNPs. Interestingly, the preference of pentameric RNPs formed by reduced N:G215C is absent after oxidation. $N_\lambda$* exhibits the largest population of hexamers observed in any sample, in agreement with the highest s-value of the reaction boundary in SV. As indicated by the time course of RNP dissociation after tenfold dilution (*Figure 6C*), oxidation enhances the average molecular weight of the RNPs but not their lifetime. We conclude that disulfide-linked N-protein tetramers can incorporate into and aid in the formation of RNPs, modulating their preferred oligomeric state, and that the cumulative effect of the N-arm mutation P13L and the LRS cysteine persists for disulfide-linked protein.

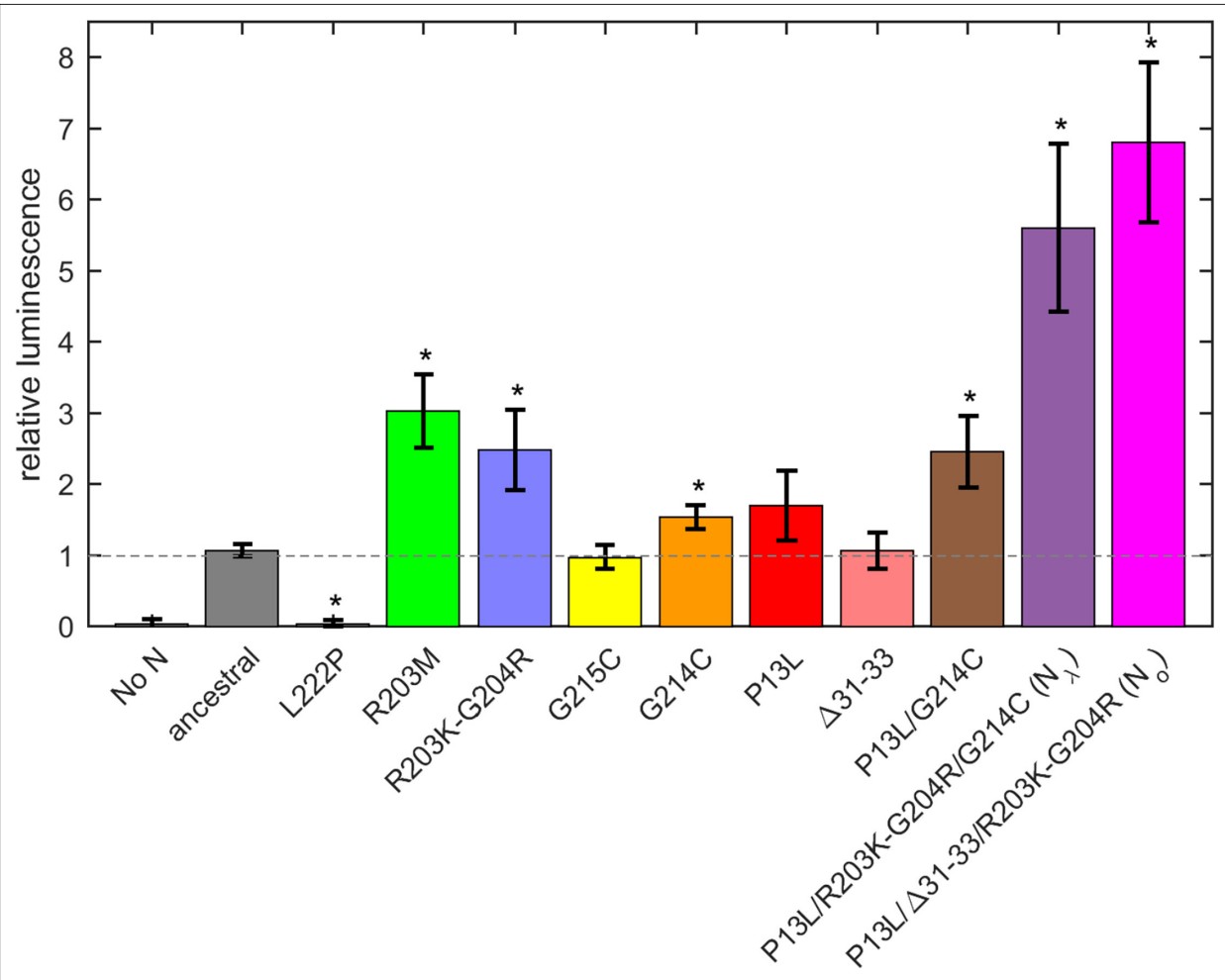

**Figure 7.** Mutation effect on packaging and cell entry in a VLP assay. Error bars are standard deviations from n=4. Stars indicate significance (p>0.95) of a two-sided Kolmogorov-Smirnov test comparing the control ancestral measurements with mutants.

## Virus-like particle formation and infectivity

We asked whether the observed modulation of RNP size and stability by N mutations impacts the formation and infectivity of virus-like particles (VLPs; *Syed et al., 2021*; *Zhao et al., 2024*). In this assay, producer 293T cells are co-transfected with plasmids for the four structural proteins S, E, M, and N of SARS-CoV-2, alongside a plasmid containing the viral packaging sequence T20 (nt 20080–22222, located near the 3' end of ORF1ab) with a luciferase reporter gene, with a total length of 4127 nt. This leads to the assembly of VLPs, which are collected from the supernatant and applied to receiver cells transfected with entry factors ACE2 and TMPRSS2. Infected receiver cells then express the luciferase reporter, and luminescence is measured as an indicator for the combined efficiency of protein expression, VLP assembly in the producer cells, and entry into the receiver cells (*Syed et al., 2021*).

First, a control experiment was carried out with the N:L222P mutant previously shown to abrogate LRS oligomerization and RNP formation. Similar to a second control without N-protein plasmid, N:L222P produced very little luminescence relative to ancestral N (*Figure 7*). A positive control was N:R203M (green), which was previously shown to significantly enhance the signal of the VLP assay (*Syed et al., 2021*), and did so in the present VLP experiments. Similarly consistent with previous reports (*Syed et al., 2021*; *Wu et al., 2021*), R203K/G204R (blue) led to increased VLP signals. Of the mutations related to N-protein binding interfaces examined in the present work, neither P13L (red), nor Δ31-33 (light red), nor G215C (yellow) alone led to significant enhancement; only G214C (orange) produced a small but statistically significant enhancement. However, the combination of the latter with P13L in P13L/G214C (brown) increased the gain, and further incorporation of the R203K/G204R double mutation to produce the full set of $N_\lambda$ mutations (purple) substantially increased the measured luminescence. A similar observation was made for the combination of P13L/Δ31-33/R203K/G204R constituting the full set of $N_o$ mutations (magenta). As expanded on in the *Discussion*, the failure to observe enhancement by P13L alone may be related to limitations of the VLP assay in sensitivity, including the restriction to a single round of infection and protein expression levels.

## P13L mutation in N-protein promotes SARS-CoV-2 replication in cell lines

Finally, we characterized the fitness of P13L mutation by introducing it into a recombinant SARS-CoV-2 reporter virus expressing an mCherry gene fused to N *via* a P2A linker (*Ye et al., 2020a*). We infected Vero-TMPRSS2 and A549-ACE2 cells at an MOI of 0.01 with Wuhan-Hu-1 (ancestral) and mutant viruses and measured mCherry fluorescence as well as viral release into the supernatant at indicated time points post-infection. This revealed that the P13L mutant exhibited a stronger mCherry

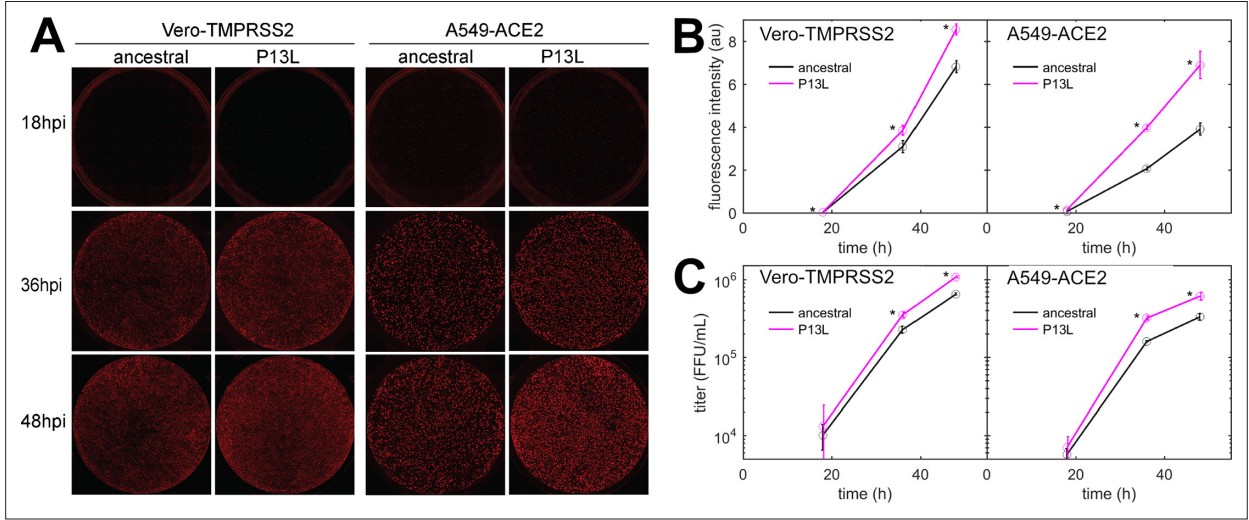

**Figure 8.** Replication kinetics of recombinant SARS-CoV-2 reporter viruses in cell lines. (**A**) Representative images of Vero-TMPRSS2 and A549-ACE2 cells infected with SARS-CoV-2 P13L or WT at different time points post-infection. (**B**) Quantification of fluorescence intensity from P13L and WT virus infections shown in (**A**). (**C**) Viral titers in the supernatant from infected cells. Error bars are standard deviations (n=3), and stars indicate significant differences on a p=0.95 confidence level.

signal throughout the course of infection (*Figure 8A and B*) and generated more progeny virus strain in both cell lines (*Figure 8C*). These findings are consistent with P13L providing a fitness advantage over ancestral virus.

## Discussion

Rapid evolution of protein binding interfaces has frequently been observed in viral protein complexes, notably in the virus-host interface, including viral surface glycoproteins as well as ribonuclear proteins and non-structural proteins, with fitness advantages being accomplished, for example, through reshaping the binding interfaces, modulating protein structural dynamics, or altering physicochemical properties (*Barozi et al., 2022*; *Evseev and Magor, 2021*; *Focosi et al., 2024*; *Planchais et al., 2022*; *Rochman et al., 2022*). Also, entirely new interactions can arise through the viral mimicry of eukaryotic short linear motifs as a result of frequent mutations in the viral protein intrinsically disordered regions, which can greatly augment the virus-host interface (*Davey et al., 2015*; *Schuck and Zhao, 2023*). The mutations we have studied here are of a different category, impacting the interactions among viral proteins that enhance viral multi-protein complexes. Previous examples include the intra-host diversity of polymerase subunit interfaces in H5N1 influenza viruses (*Welkers et al., 2019*). While such mutations are not directly targeted towards the host, they may still contribute to host adaptation or be balancing other mutation effects in an epistatic network, conceivably involving modulation of local

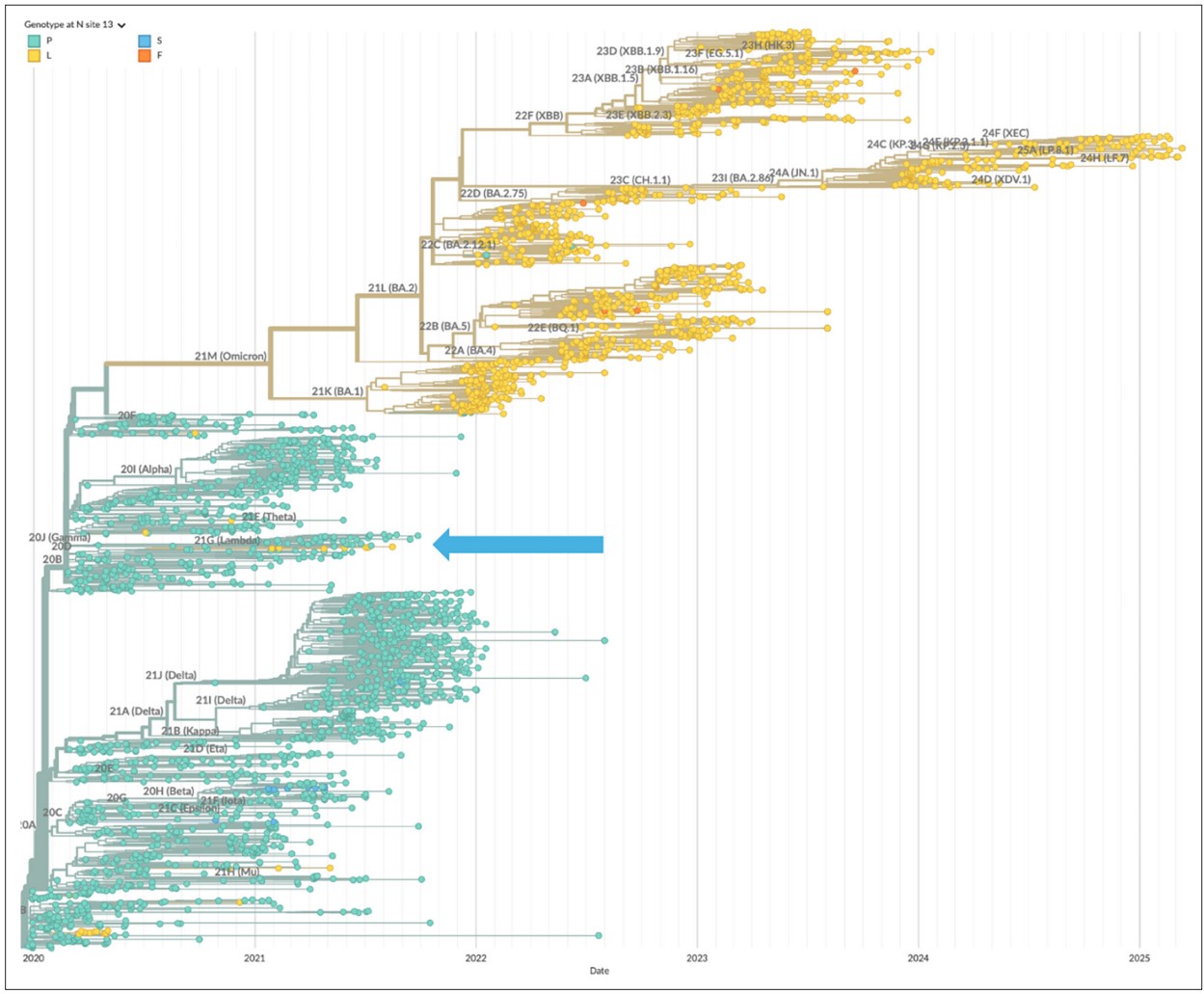

**Figure 9.** Mutations of N:P12 across the phylogenetic tree of SARS-CoV-2. Shown are all-time global sequence samples with clade labels and color-coded amino acid at position 13, with the ancestral P13 in green and P13L in yellow. The blue arrow points to the Lambda sequences. Additionally, a cluster of P13L mutations occurred in India in clade 19 A. The phylogenetic tree was generated by Nextstrain (*Hadfield et al., 2018*).

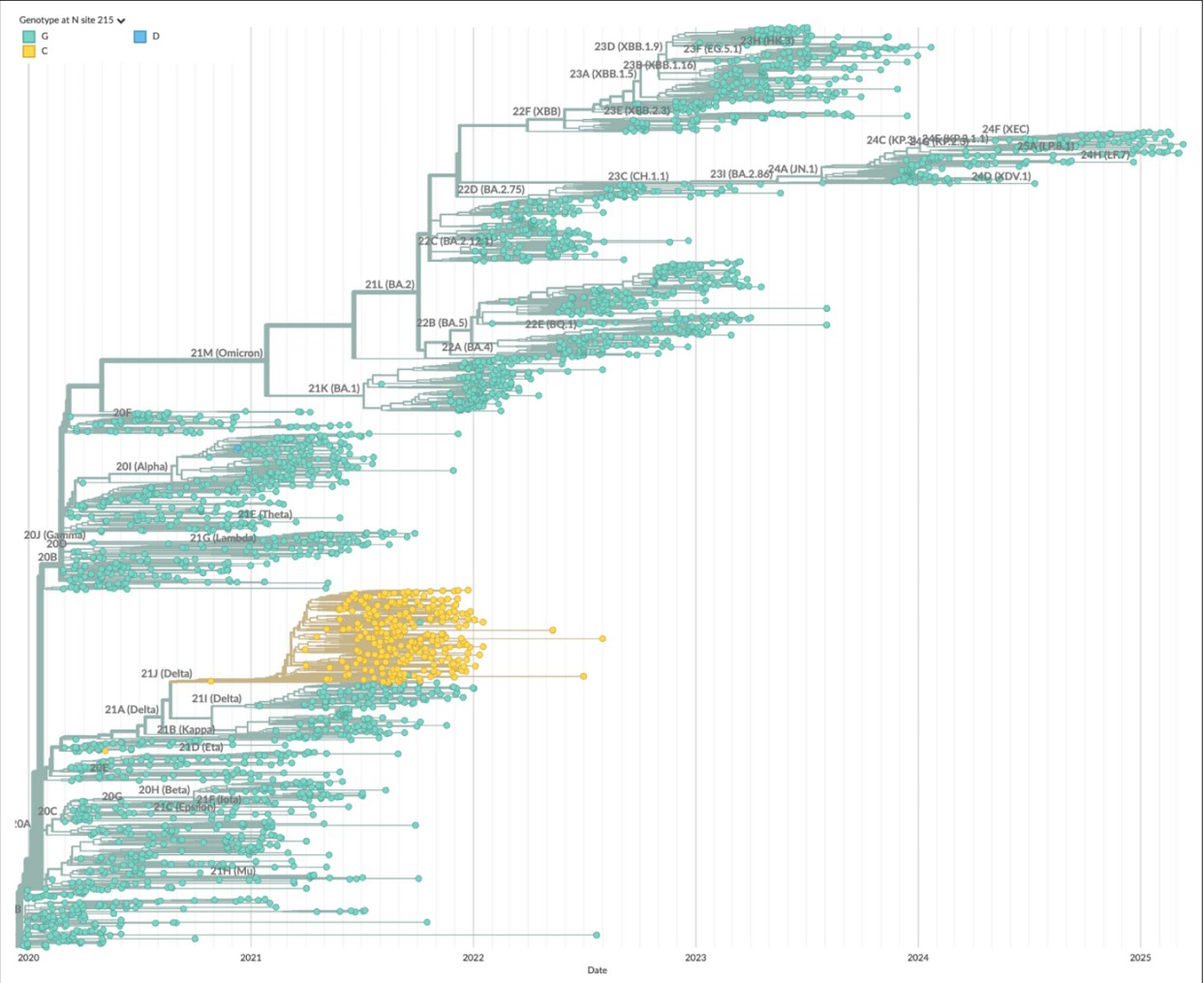

**Figure 10.** Mutations of N:G215 across the phylogenetic tree of SARS-CoV-2. Shown are all-time global sequence samples with clade labels and color-coded amino acid at position 215, with the ancestral G215 in green and G215C in yellow. The phylogenetic tree was generated by Nextstrain (*Hadfield et al., 2018*).

effective protein concentrations (*Li et al., 2023b*). Irrespective of their complete context, they can provide valuable insights into viral protein mechanisms.

Specifically, we have described three different mutations of SARS-CoV-2 N-protein that, in convergent evolution, strengthen the formation of RNPs and enhance viral assembly. N:G214C, N:G215C, and N:P13L have been independently introduced (as highlighted in the phylogenetic trees *Figure 9*; *Figure 10*; *Figure 11*), and persisted in the defining set of mutations in their respective variants of concern (Lambda, Delta, and Omicron, respectively). We have shown here that N:P13L confers a fitness advantage in cell lines, and similarly, N:G215C was shown by *Kubinski et al., 2024* to impart improved viral growth. This correlates well with our results studying their molecular mechanisms.

For both of the cysteine mutants, molecular dynamics simulations and biophysical studies show how cysteines augment self-association interfaces by extending and redirecting the transiently formed helical coiled-coils in the intrinsically disordered LRS, which play a central role in the assembly of RNPs. By contrast, for N:P13L, unexpectedly, the evolution of RNP stability goes beyond modulation of a previously existing binding interface, and instead, we observe the de novo formation of an additional dynamic self-association interface in the distant disordered N-arm through the stabilization and stacking of transient β-sheets, that we hypothesize cooperatively contributes to the stability of RNPs. Even though the solution affinity of the N-arm P13L interface is ultra-weak, the average local concentration of N-arm chains across the RNP volume (in a back-of-the-envelope calculation assuming a

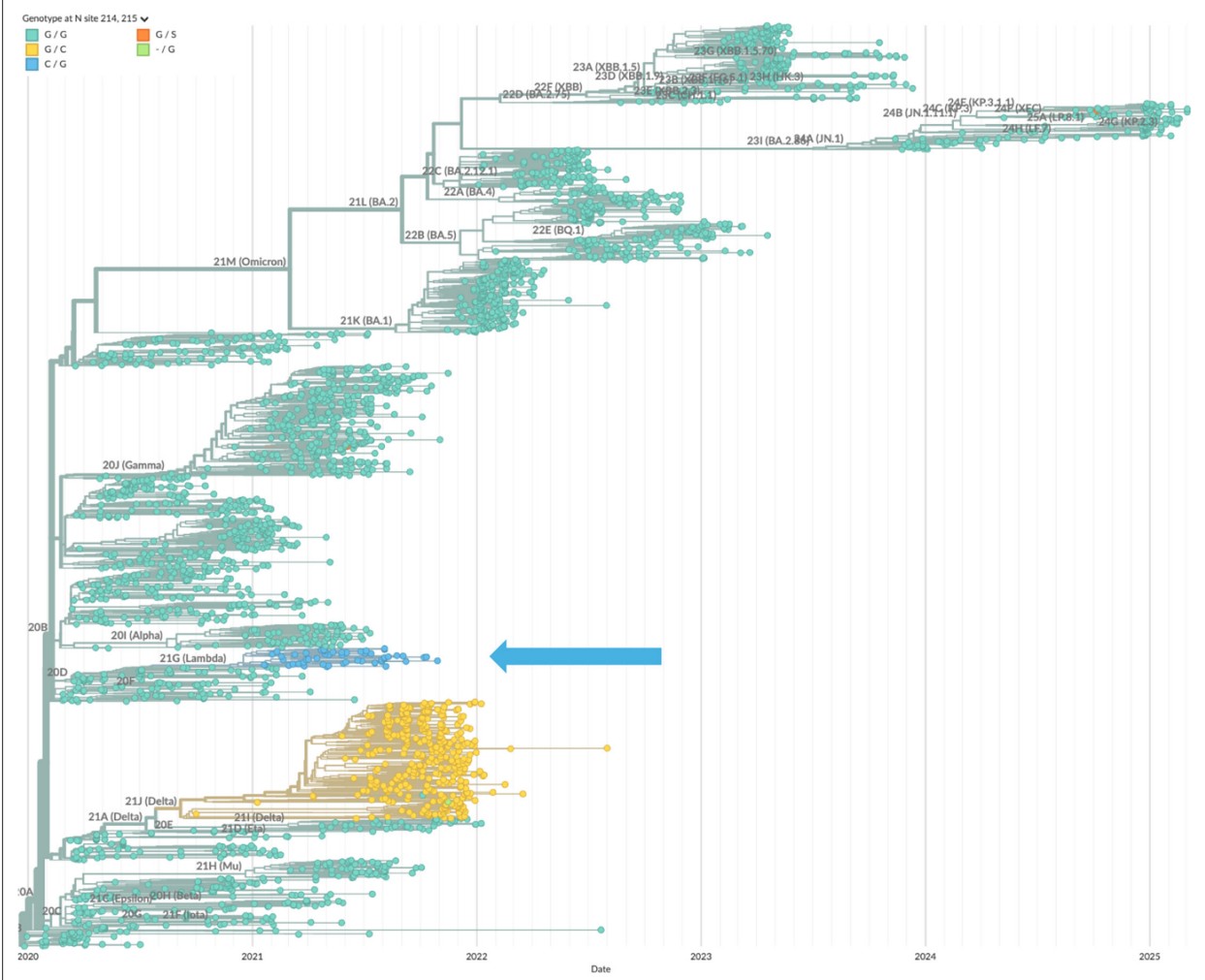

**Figure 11.** Mutations of N:G214 and N:G215 across the phylogenetic tree of SARS-CoV-2. Shown are all-time sequence samples in South America with clade labels and color-coded amino acid at position 214 and 215. The combination of 214 C/G215 strain 21 G (Lambda) is shown in blue, whereas the combination G214/215 C of strain 21 J (Delta) is shown in yellow. The phylogenetic tree was generated by Nextstrain (*Hadfield et al., 2018*).

≈14 nm cube *Klein et al., 2020* with a dodecameric N cluster) is ≈7.4 mM, such that disordered N-arm peptides could well create populations of N-arm clusters stabilizing RNPs through this interface.

However, besides the RNP-stabilizing mutants, we have also observed unexpected RNP destabilization by the ubiquitous R203K/G204R double mutation, which may be caused by the introduction of additional charges close to the self-association interface in the LRS. In our experiments, this destabilization is more than compensated for by the P13L mutation. (Another scenario where ultra-weak interactions can have a critical impact is in molecular condensates. We previously reported the suppression of LLPS by the R203K/G204R mutation, which is rescued by the additional P13L/Δ31–33 mutation (*Nguyen et al., 2024*). This is consistent with compensatory weak stabilizing and destabilizing impacts of weak interactions on the RNP observed here.)

We arrive at a picture of SARS-CoV-2 RNPs that is far from structurally well defined, matching the concept of fuzzy complexes (*Wu and Fuxreiter, 2016*). On a molecular level, large portions of the SARS-CoV-2 N-protein (the N-arm, C-arm, and linker) are intrinsically disordered and highly flexible (*Cubuk et al., 2021*; *Różycki and Boura, 2022*), which persists in the presence of bound nucleic acid (*Cubuk et al., 2024*; *Guseva et al., 2021*; *Schiavina et al., 2022*). It appears that conformational freedom is also retained to a significant degree in the RNPs. This flexibility could be advantageous for accommodating various RNA secondary structures (*Carlson et al., 2022*; *Landeras-Bueno et al., 2025*) and favorably balance the energetic cost of RNP disassembly that is required immediately after viral entry. Also, this serves to accommodate significant sequence variation (*Davey et al., 2011*;

*Duro et al., 2015*; *Schuck and Zhao, 2023*). SARS-CoV-2 RNPs appear highly heterogeneous in EM (*Carlson et al., 2022*; *Landeras-Bueno et al., 2025*; *Yao et al., 2020*), and this is reflected in the polymorphic oligomeric states of RNP species we observe in SV-AUC and MP, that we believe is driven by promiscuous self-association or clustering of transient LRS helices (*Zhao et al., 2022*). Extending previously described characteristics of fuzziness in protein complexes (*Duro et al., 2015*; *Fuxreiter, 2018*; *Tompa and Fuxreiter, 2008*), plasticity seems to involve even basic architectural principles, considering not only the emergence of new distant stabilizing interfaces such as described here in the N-arm, but also the possibility of RNP assembly of truncated $N_{210-419}$* lacking one of the major nucleic acid binding interfaces (*Adly et al., 2023*; *Bouhaddou et al., 2023*; *Mears et al., 2025*; *Mulloy et al., 2025*; *Syed et al., 2024*) (see below).

Unfortunately, this intrinsic heterogeneity poses significant methodological hurdles. Nonetheless, salient structural features and assembly principles may be derived from constraints of known binding interfaces and oligomeric states of the RNP and its subunits, as observed in SV-AUC and MP. While the arrangement sketched in *Figure 1C* satisfies these requirements, alternate less symmetrical configurations can be conceived that seem at least equally likely and may coexist in polydisperse mixtures of RNPs. For example, there is no evidence to exclude the possibility of anti-parallel LRS helices pointing the folded nucleic acid -binding domains in different relative orientations, or of mixed co-assemblies with $N_{210-419}$* subunits lacking the NTD (*Figure 1—figure supplement 1*). Uniformity of N-protein/RNA clusters may not be relevant for adequate gRNA condensation.

Beyond the structural model, to study the effect of a larger number of N-protein mutations derived from variants of concern side-by-side in the context of virus assembly, we have carried out experiments using a VLP assay (*Syed et al., 2021*; *Figure 7*). In these experiments, all four structural proteins are transfected into 293T cells to package a reporter RNA into VLPs, and their infection of receiver cells can be compared. While this assay has been widely used for rapid assessment of spike protein and N variants (*Syed et al., 2021*), it has limitations due to the addition of non-genomic RNA and the lack of double membrane vesicles from which gRNA emerges through the NSP3/NSP4 pore complex potentially poised for packaging (*Bessa et al., 2022*; *Ke et al., 2024*; *Ni et al., 2023*). It should also be recognized that the results do not directly reflect the relative efficiency of RNP assembly only, since protein expression levels, their localization, and their posttranslational modifications are not controlled for. Susceptibility to such factors might be exacerbated with mutations that modulate weak protein interactions. For example, as shown previously (*Syed et al., 2024*; *Zhao et al., 2024*), a GSK3 inhibitor inhibiting N-protein phosphorylation significantly enhances VLP formation and eliminates the advantage provided for by the N:G215C mutation relative to the ancestral N – presumably due to an increase in assembly-competent, non-phosphorylated N-protein erasing an affinity advantage. A similar process may be underlying the absent or marginal improvement in VLP readout from the cysteine LRS mutants and P13L at the achieved transfection level in the present work, and the enhanced signal from R203K/G204R and R203M (the latter being consistent with previous reports *Li et al., 2025*; *Syed et al., 2021*) modulating protein phosphorylation. Nonetheless, mirroring the results of the biophysical in vitro experiments, the addition of RNP-stabilizing P13L and G214C mutations on top of R203K/G204R led to a significantly larger VLP signal.

The VLP assay may also be limited in sensitivity to mutation effects due to its restriction to a single round of infection. To avoid this and other potential limitations of the VLP assay for the study of viral packaging, for the key mutation N:P13L, we carried out reverse genetics experiments. These showed the sole N:P13L mutation significantly increases viral fitness (*Figure 8*).

Regarding the cysteine mutations that have been repeatedly introduced in the LRS prior to the rise of the Omicron variants of concern, it is an open question whether they lead to covalent bonds in vivo or in the VLP assay. While examples of disulfide-linked viral nucleocapsid proteins have been reported (*Kubinski et al., 2024*; *Prokudina et al., 2004*; *Wootton and Yoo, 2003*), a methodological difficulty in their detection is artifactual disulfide bond formation post-lysis of infected cells (*Kubinski et al., 2024*; *Wootton and Yoo, 2003*). However, our results clearly show that a major effect of the cysteines already arises in reduced conditions without any covalent bonds, through extension of the LRS helices and concomitant redirection of the disordered N-terminal sequence. While oxidized tetrameric N-proteins of N:G214C and N:G215C can be incorporated into RNPs, the covalent bonds provided only marginally improved RNP stability. Interestingly, the introduction of cysteines imposes preferences of RNP oligomeric states dependent on oxidation state, consistent with our MD simulations highlighting

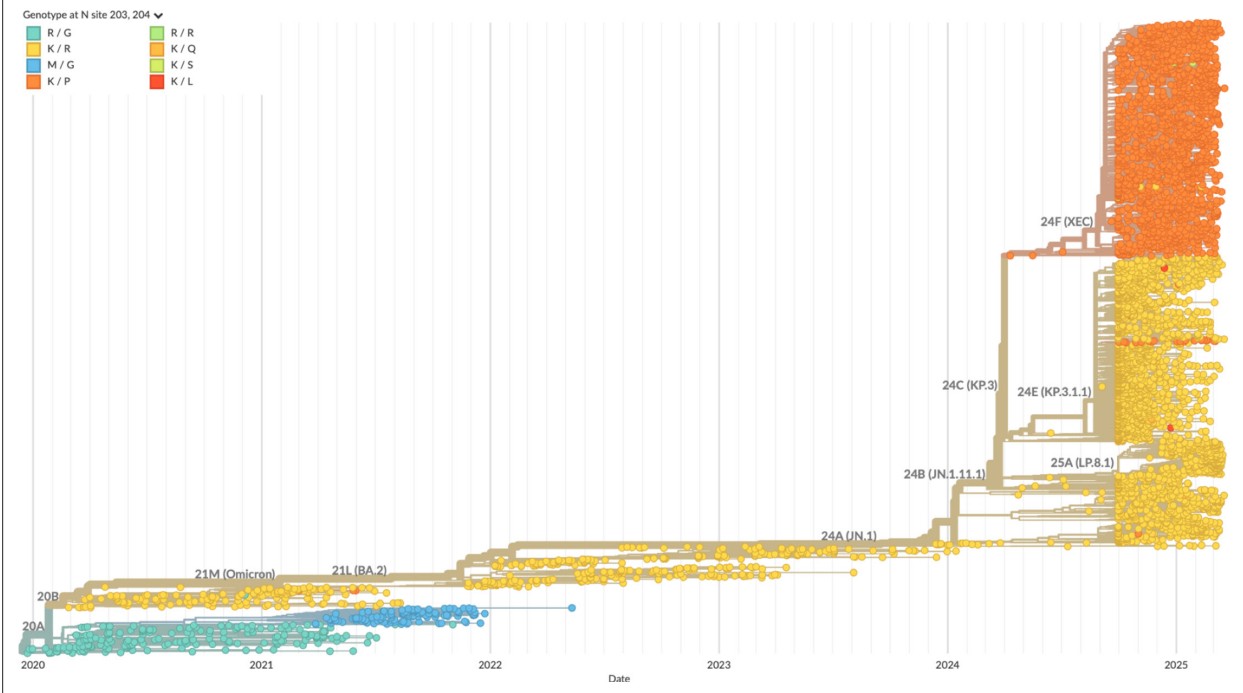

**Figure 12.** Mutations of N:R203 and N:G204 across the phylogenetic tree of SARS-CoV-2. Shown are global sequence samples mostly representing sequences of the recent 6 months, with clade labels and color-coded amino acid at positions 203 and 204. The ancestral combination of R203/G215 is shown in green, the mutation 203 M of the Delta VOC in blue, the combination 203 K/204 R common to Alpha and Omicron VOCs in yellow, and the combination 203 K/204 P defining in the Omicron XEC variant in orange. The phylogenetic tree was generated by Nextstrain (*Hadfield et al., 2018*).

the impact of cysteine orientation of 214 C *versus* 215 C relative to the hydrophobic surface of the LRS helices. Overall, considering potentially detrimental structural constraints from covalent bonds on LRS clusters seeding RNPs, energetic penalties on RNP disassembly, as well as the required monomeric state of the LRS helix for interaction with the NSP3 Ubl domain (*Bessa et al., 2022*), at present, it is unclear to what extent the formation of disulfide linkages between LRS helices would be beneficial or detrimental in the viral life cycle.

Recent work by the Soranno laboratory has identified an additional function of the disordered N-arm in transiently interacting with the NTD (*Cubuk et al., 2021*) and dynamically enhancing the affinity of the NTD for RNA (*Cubuk et al., 2024*). Using single-molecule Förster Resonance Energy Transfer (smFRET), a fourfold modulatory effect of the P13L/Δ31-33 mutation on the NTD RNA binding affinity was observed in N-arm-NTD constructs. Through MD simulations, the reduced NTD affinity for RNA was attributed to the N-arm Δ31-33 deletion (*Cubuk et al., 2024*). Superficially, this may seem in slight conflict with our results of similar $T_{10}$ affinity of full-length ancestral N with and without the P13L/Δ31-33 mutation, but results were obtained in different buffer conditions (50 mM TRIS, pH 7.4 in *Cubuk et al., 2024 versus* 20 mM HEPES, 150 mM NaCl, pH 7.5 in the present work). In any event, RNA binding of NTD and stabilization of the RNP are different processes; any modulation of N-arm contributions to NTD-RNA interactions through Omicron N-arm mutations Δ31-33 may coexist and be over-compensated for by N-arm self-association interfaces through P13L modulating RNP subunit interactions in the high local N-arm density of the RNP.

The double mutant R203K/G204R arose early in the pandemic and was adopted in several variants of concern (including Alpha, Gamma, Lambda, Zeta, and Omicron BA.1) with the triple nucleotide changes G28881A, G28882A, and G28883C (*Mears et al., 2025*; *Syed et al., 2024*; *Figure 12*). As mentioned above, on the protein level, N:R203K/G204R has been shown to alter phosphorylation (although in different ways in in vitro VLP or in vivo reverse genetics experiments; *Johnson et al., 2022*; *Syed et al., 2024*; *Yun et al., 2022*), and phosphorylation in turn reduces nucleic acid binding and promotes viral replication as opposed to assembly functions (*Botova et al., 2024*; *Bouhaddou et al., 2023*; *Carlson et al., 2020*; *Syed et al., 2024*). Adding to such a switch, in the present work, we observed the loss of RNP stability of N:R203K/G204R relative to the ancestral N, extending the

previous observation of reduced LLPS propensity of N:R203K/G204R (*Nguyen et al., 2024*). Simultaneously, on the RNA level, the N:R203K/G204R mutations also lead to the new formation of a TRS sequence ACGAAC underlying the expression of $N_{210-419}$* in virus-infected cells (though not expected to occur with N:R203K/G204R in the VLP assay lacking the viral RNA-dependent RNA polymerase). It has been hypothesized that $N_{210-419}$* confers increased viral fitness through the suppression of the host anti-viral response (*Mears et al., 2025*; *Mulloy et al., 2025*), and that it can assist RNP formation (*Bouhaddou et al., 2023*; *Syed et al., 2024*). However, the contribution of $N_{210-419}$* to assembly is still unclear: although it is remarkably capable of forming RNPs in vitro and VLP assays (*Adly et al., 2023*; *Bouhaddou et al., 2023*; *Syed et al., 2024*), in infected cells and virions, $N_{210-419}$* has been detected only as a minority species (*Mears et al., 2025*; *Mulloy et al., 2025*). Also, the recent major Omicron XEC variant (*Scarpa et al., 2025*; which had close to 60% global frequency at the beginning of 2025; *Figure 12*) exhibits a fourth consecutive nucleotide change G28884C that maintains a similar RG mutation forming R203K/G204P but ablates the canonical TRS sequence, such that continued expression of $N_{210-419}$* in XEC is in question. We propose that an alternative or additional mechanism to retain viral assembly functions may be presented by the accompanying P13L mutation, which our data suggest can more than restore loss of RNP stability in the combination of RG mutations with P13L. This combination occurs in $N_o$ and all Omicron variants so far and was even further stabilized with a cysteine in the LRS in $N_\lambda$.

In conclusion, it has been proposed that mutations in SARS-CoV-2 N protein that affect viral assembly can impact infectivity and fitness (*Bouhaddou et al., 2023*; *Syed et al., 2024*; *Wu et al., 2021*; *Zhao et al., 2022*). We believe the observed modulations of the RNP assembly and stability studied here highlight a key mechanism for this. Although effects on fitness of viruses carrying N mutations are most likely multi-factorial, they have been observed in reverse genetics tissue culture experiments previously for N:R203K/G204R (*Johnson et al., 2022*; *Mears et al., 2025*; *Wu et al., 2021*), N:G215C (*Kubinski et al., 2024*), and in the present work for N:P13L. On the other hand, the rise of new variants of concern was usually dominated by their spike protein mutations (with the exception of 21I replacement by 21J which has identical spike mutations but acquired N:G215C *Marchitelli et al., 2021*; *Stern et al., 2021*; *Zhao et al., 2022* in the rise of Delta variant), and in many cases, N mutations of previously dominant variants were completely replaced by another set of N mutations (dramatically exemplified in the displacement of Delta by Omicron variants). This reinforces the view that these N mutations are secondary to alterations in the immune landscape and transmissibility as the primary driver of evolution (*Markov et al., 2023*). Nonetheless, the remarkable plasticity of RNPs offers multiple avenues to modulate stability and to compensate for potentially RNP-destabilizing effects of mutations that are beneficial in other ways. In convergent evolution, this has been a constant theme of N protein mutations throughout the SARS-CoV-2 pandemics up until today. We hypothesize that the 'fuzziness' and pleomorphic ability of RNP assembly, with its variable distribution of overall binding energy into several different weak or ultra-weak protein interfaces, and the poor structural definition ranging from flexible chain configurations to polydisperse oligomeric states, provides an evolutionary advantage of orchestrated disorder to promote epistatic interactions and facilitate host adaptation.

## Materials and methods

**Key resources table**

| Reagent type (species) or resource | Designation | Source or reference | Identifiers | Additional information |
|---|---|---|---|---|
| Cell line (*Chlorocebus sabaeus*) | Vero E6-TMPRSS2 | JCRB cell bank | JCRB1819 | |
| Cell line (*Homo sapiens*) | A549-hACE2 | BEI | NR-53522 | |
| Cell line (*Mesocricetus auratus*) | BHK21-ACE | *Li et al., 2023a* | | |
| Cell line (*Homo sapiens*) | 293T | ATCC | RRID:CVCL_1926 | |
| Strain, strain background (*Escherichia coli*) | BL21(DE3)pLysS | Thermo Fisher | C606003 | |

*Continued on next page*

*Continued*

| Reagent type (species) or resource | Designation | Source or reference | Identifiers | Additional information |
|---|---|---|---|---|
| Recombinant DNA reagent | pLVX-EF1alpha-SARS-CoV-2-E-2xStrep-IRES-Puro | Addgene | RRID:Addgene_141385 | |
| Recombinant DNA reagent | pLVX-EF1alpha-SARS-CoV-2-M-2xStrep-IRES-Puro | Addgene | RRID:Addgene_141386 | |
| Recombinant DNA reagent | pLVX-EF1alpha-SARS-CoV-2-N-2xStrep-IRES-Puro | Addgene | RRID:Addgene_141391 | |
| Recombinant DNA reagent | pET29a(+) (plasmid) | Genescript | | |
| Recombinant DNA reagent | pIRES2-EGFP | NovoPro | V011106 | |
| Sequence-based reagent | N1-43 (N-arm) and N210-246 | ABI Scientific | | For sequences see *Supplementary file 1* |
| Sequence-based reagent | T10, SL7 | Integrated DNA Technologies | | For sequences see *Supplementary file 1* |
| Software, algorithm | SEDFIT | Biophys. J. (2000) 1606–1619 | RRID:SCR_018365 | Can be retrieved from https://doi.org/10.7910/DVN/4JPARC |

## Protein expression and purification

Full-length N-protein of the wild type and mutant SARS-CoV-2 were expressed and purified as described previously (*Nguyen et al., 2024*; *Zhao et al., 2024*). Briefly, One Shot BL21(DE3)pLysS *E. coli* (Thermo Fisher Scientific, Carlsbad, CA) was transformed using a pET29a(+) plasmid vector, which contains a kanamycin-resistant gene and the gene encoding the N-protein of interest preceded by 6xHis tag followed by a Tobacco Etch Virus (TEV) cleavage site at the N-terminus. After cell lysis, the protein was purified by $Ni^{2+}$ affinity chromatography, where on-column unfolding by urea and refolding steps were carried out to remove protein-bound bacterial nucleic acid (*Carlson et al., 2020*). After tag cleavage by TEV protease, 6xHis tag removal was verified through another round of affinity chromatography and/or *via* mass spectrometry. Cleaved protein was subjected to a final size exclusion chromatography followed by dialysis into working buffer (20 mM HEPES, 75 mM NaCl, pH 7.50, supplemented with 1 mM TCEP for cysteine-containing proteins, unless otherwise mentioned). Protein purity was validated by SDS-PAGE and the absence of nucleic acid was confirmed by an absorbance ratio 260/280 of ≈0.50–0.55. Final protein concentration was determined by UV-Vis spectrophotometry or by refractive index detected SV-AUC.

N peptides were purchased from ABI Scientific (Sterling, VA), purified by HPLC, examined by matrix-assisted laser desorption/ionization for purity and identity, and lyophilized. The oligonucleotide $T_{10}$ and stem–loop RNA SL7 were purchased from Integrated DNA Technologies (Skokie, IL) and purified by HPLC and lyophilized by the vendor. After reconstitution, SL7 was subject to thermal denaturation at 95 °C for 2 min followed by gradual cooling to room temperature over 1–2 hr. For sequences of oligonucleotides and peptides, see *Supplementary file 1*.

The 5,5'-Dithiobis-(2-Nitrobenzoic Acid) (DTNB; catalog #22582) and Cysteine-HCl (catalog #44889) were purchased from Thermo Fisher Scientific Inc (Waltham, MA). Free thiols in the protein samples were quantified using the Ellman's assay by following the standard protocol from the vendor. Briefly, free thiols react with DTNB, resulting in a measurable yellow-colored product, TNB. The quantity of sulfhydryl groups was calculated by the absorbance of the sample using the molar extinction coefficient of TNB at 412 nm ($14,150\ M^{-1}cm^{-1}$). The same samples were analyzed by SDS-PAGE without the addition of a reducing agent. The relative intensities of the monomer and dimer bands reflect the amounts of the reduced and oxidized forms, respectively.

## Structure prediction and MD simulations

In the studies of LRS peptides, the initial structures of the monomer and oligomers of the ancestral $N_{210-246}$ were predicted using AlphaFold3 (AF3). The predicted conformations agreed with those reported in *Zhao et al., 2023*; *Zhao et al., 2022*, with the oligomers showing a parallel, left-handed

coiled-coil signature. Point mutations were introduced by replacing the corresponding glycine residue with cysteine. Graphics were created using ChimeraX.

MD simulations were performed for monomers and trimers of the ancestral peptides $N_{210-246}$ and two mutant sequences (G214C and G215C), following the protocol described in *Zhao et al., 2023*. In each case, the initial structure was the top-ranked AF3 model. Each simulation was extended for 250 ns after thermal equilibration under experimental conditions (T=20 °C, *p*=1 atm, pH 7.5, and 75 mM NaCl) using the isothermal-isobaric ensemble as implemented in NAMD. The all-atom representation of the CHARMM (param36) force field was used. Structural stabilization of the helical regions was observed within a few nanoseconds, and data were analyzed over the last 200 ns of the simulations.

Structure of the N-arm oligomers was predicted using ColabFold (*Mirdita et al., 2022*), assembling 20 copies of ancestral $N_{10-20}$, $N_{10-20}$:P13L, $N_{10-20}$:P13S, or $N_{10-20}$:P13T. Structures were analyzed and displayed using ChimeraX (*Pettersen et al., 2021*).

## Electron microscopy

Carbon-coated 200 mesh copper TEM grids were glow discharged for ≈15 s. Next, 4 µL of the sample solution was deposited onto the grids and incubated for 2 min. After incubation, excess sample solution was removed by gently touching the edge of the grids with filter paper. A large drop of distilled water was then placed on the grids for 1 min, followed by removal of the water by contacting the grid edge with filter paper. This rinsing step was repeated three times. The grids were then stained with 5 µL of 1% uranyl acetate (UA) solution for 20 s. Any excess UA was removed by touching the edge of the grids with filter paper and then left to air dry. Finally, the grids were examined with a FEI Tecnai12 Transmission Electron Microscope (FEI, Hillsboro, Oregon), operating at 120 keV beam energy. TEM images are captured using a high-speed, high-resolution Gatan Rio 3k × 3k CMOS camera (Gatan, Warrendale, PA).

## Sedimentation velocity analytical ultracentrifugation

SV-AUC experiments were conducted in a ProteomeLab XL-I analytical ultracentrifuge (Beckman Coulter, Indianapolis, IN) using standard protocols as previously described (*Schuck et al., 2015*). AUC cell assemblies filled with samples using 12- or 3 mm charcoal-filled Epon double-sector centerpieces with sapphire windows were loaded into An-50 or An-60 rotors and temperature equilibrated in the rotor chamber at 20 °C for 2–3 hr. Subsequently, radial scans were acquired with Rayleigh interference optics and absorbance optics at 260 nm and/or 280 nm. Calibration factors for the instrument were determined according to previously published methods (*Ghirlando et al., 2013*). Sedimentation boundary data were analyzed using sedimentation coefficient distribution *c(s)* model in the software SEDFIT using maximum entropy regularization (*Schuck, 2016*).

## Mass photometry

Mass photometry measurements were performed in a TwoMP instrument (Refeyn, UK) following the standard protocol (*Wu and Piszczek, 2021*) unless otherwise mentioned. Samples were loaded in the mini-wells formed by a silicone gasket which was placed on top of a coverslip mounted on the microscope stage. Two configurations of sample loading were used in the current study. For the time-dependent experiments, the samples were first prepared in the Eppendorf tubes prior to the MP experiment. Then the working buffer (9 µL) was loaded onto the coverslip for focusing the objective. Subsequently, the sample was added to the buffer droplet, gently mixed, and the measurement was initiated immediately as the 1st time point. The subsequent acquisitions for the same sample continued for specific time intervals. For the samples that were not subject to this dilution/mixing configuration, sample mixtures were equilibrated 2–3 hr, focusing was achieved by using the buffer-free option provided by the data acquisition software, and data was collected immediately after sample application. In either configuration, the impact of surface binding on the sample concentration is <1% and negligible, as described in the *Appendix*. Calibration of the TwoMP instrument was performed using the two calibrants, beta-amylase from sweet potato (Sigma A8781) and thyroglobulin from bovine thyroid (Sigma T9145) as recommended by the manufacturer. MP data was acquired with AcquireMP software and the analysis was performed with DiscoverMP software (Refeyn, UK).

Rapid mixing experiments were carried out on a OneMP-MassFluidix HC system (Refeyn Ltd., Oxford, UK). A rapid-dilution microfluidic chip (MP-CON-51001, Refeyn Ltd., Oxford, UK) was

connected to the buffer, sample, and waste lines. A sample was injected into the chip at a flow rate of 8 µL/min, while the buffer was flowed at 1100 µL/min, and 1 min videos were recorded for data acquisition.

## Circular dichroism spectroscopy

CD spectra were acquired in a Chirascan Q100 (Applied Photophysics, UK) at 20 °C. Measurements were performed in 0.1 mm (peptides) or 1 mm (proteins) pathlength cuvettes with 1 nm steps, and a 1 s integration time per data point. Each spectrum represents the average of three independent scans with background subtraction applied.

## Dynamic light scattering

Dynamic light scattering measurements of the samples were performed in a Prometheus Panta (Nanotemper, Germany) instrument at 20 °C. The samples were loaded into capillaries (Nanotemper PR-AC002) and autocorrelation functions (ACFs) were acquired using the 405 nm laser at the detection angle of 140°. The ACFs were analyzed with discrete species models, size-distribution models, and cumulant analysis in SEDFIT (*Parker and Lollar, 2021*).

## Virus-like particle assay

The plasmids pLVX-EF1alpha-SARS-CoV-2-E-2xStrep-IRES-Puro (#141385), pLVX-EF1alpha-SARS-CoV-2-M-2xStrep-IRES-Puro (#141386), and pLVX-EF1alpha-SARS-CoV-2-N-2xStrep-IRES-Puro (#141391) were obtained from Addgene. Plasmid pLuc-T20 was a kind gift from Jennifer A. Doudna. The plasmid pIRES2-EGFP (Cat # V011106) was purchased from NovoPro. Plasmids encoding ACE2, pcDNA3.1(+)-SARS-CoV-2 WA1-S, and pGAGGS-TMPRSS2 were previously described (*Shi et al., 2022*; *Shi et al., 2021*). To construct the plasmid pcDNA3.1(+)-N, the N gene was cloned into pcDNA3.1(+) between the BamHI and NotI sites. Mutations in N were generated by QuikChange site-directed mutagenesis and verified by whole plasmid sequencing. Similarly, to prepare plasmid pIRES2-ME, the M gene was first cloned into pIRES2-EGFP between NheI and BamHI sites, followed by the insertion of the E gene into the resulting plasmid between NotI and a PvuI site introduced after the start codon of EGFP.

The SARS-CoV-2 virus-like particles (VLPs) were prepared as previously described (*Syed et al., 2021*; *Zhao et al., 2024*) with some modifications. HEK 293T cells were acquired from ATCC (CRL-11268; RRID:CVCL_1926, mycoplasma tested and authenticated by STR DNA profiling) immediately prior to SC2-VLP production. $0.8 \times 10^6$ 293T cells per well were plated in a six-well plate and allowed to grow 24 hr before transfection. Plasmids Cov2-N (1.34), CoV2-M-IRES-E (0.659), CoV-2- Spike (0.0032) and Luc-T20 (2.0) at indicated mass ratios for a total of 4 µg of DNA were diluted in 250 µL Opti-MEM reduced serum medium (Gibco cat# 31985062) at room temperature. 12 µL of TransIT–293 Transfection Reagent (Mirus Bio, cat# MIR 2704) equilibrated to room temperature was added to each sample, and immediately vortexed. The transfection mixtures were incubated for 25 min at room temperature and then added dropwise to 293T cells after media change with 2 mL of DMEM containing fetal bovine serum and penicillin/streptomycin. Media was changed after 24 hr of transfection and at 48 hr post-transfection, VLP containing supernatant was collected and filtered using a 0.22 µm syringe filter. Cells in each well were rinsed with 1 mL of PBS and then lysed directly in the wells with 300 µL of NuPAGE LDS Sample Buffer (Invitrogen, cat# NP0008) containing HALT protease inhibitors (Thermo Fisher Scientific, cat# 87786) and 10 mM DTT.

For the luciferase assay, the wells in 12-well plates were pre-treated with 1 mL poly-D- Lysine (Thermo Fisher Scientific, cat# A3890401) for 30 min, which was removed, washed with PBS once, and finally, the plates were airdried. In each well of these pre-treated plates, $1.5 \times 10^5$ receiver cells (293T cells transfected with the plasmids encoding ACE2 and pGAGGS-TMPRSS2 at a mass ratio of 1:1 using the TransIT–293 transfection reagent) were plated. The next day, the media was replaced, and the cells were infected with 250 µL of supernatant from the producer cells. After 24 hr, the media was removed, and cells were rinsed with PBS and lysed in 150 µL passive lysis buffer (Luciferase Assay System, Promega, cat# E1500) for 15 min at room temperature with gentle rocking. 20 µL of each lysate was transferred to an opaque black 96-well plate in triplicate, and 50 µL of reconstituted luciferase assay buffer was added and mixed with each lysate. Luminescence was measured immediately after mixing using a TECAN plate reader.

## Generation of recombinant SARS-CoV-2

The generation and use of recombinant SARS-CoV-2 (rSARS-CoV-2) viruses in tissue culture at biosafety level 3 were approved by NIH Institutional Biosafety (IBC) and the Dual Use Research of Concern Institutional Review Entity (DURC-IRE) Committees (IBC approved case number: RD-22-XI-11).

We generated rSARS-CoV-2 viruses using a bacterial artificial chromosome (BAC)-based SARS-CoV-2 reverse genetics system (*Ye et al., 2020a*), with detailed construction and recovery procedures described in our previous study (*Li et al., 2023a*). Specifically, we fused the mCherry gene with the N gene via a 2 A linker and created an intermediate plasmid, pUC57-NEM, containing a portion of the pBAC-SARS-CoV-2 genome digested with BamHI and SalI restriction enzymes. The P13L mutation was introduced via site-directed mutagenesis on the pUC57-NEM plasmid. PCR amplification was performed to generate fragments containing the NEM genes, which were then assembled with the BamHI/SalI-digested larger fragment using the NEBuilder HiFi DNA Assembly Master Mix. A furin cleavage site mutation (R685S) served as the backbone for introducing mutations in the N protein.

Plasmids were purified using the QIAGEN Plasmid Maxi Kit. Confluent BHK21-ACE2 cells ($2\times10^6$ cells/well in 6-well plates, in duplicate, RRID:CVCL_1914, ATCC) were transfected with 2.5 μg/well of pBAC-SARS-CoV-2 using the TransIT-LT1 transfection reagent. After 6 hr, the medium was replaced with fresh DMEM containing 2% FBS. At 48–72 hr post-transfection, mCherry-positive cells exhibiting signs of viral infection were detached, collected with the supernatant, labeled as P0, and stored at −80 °C. The P0 viral stock was centrifuged to remove cell debris and used to infect fresh Vero E6-TM-PRSS2 (RRID:CVCL_0574, ATCC) cells for 48–72 hr. The resulting P1 viral stock was collected, and viral titers were determined following next-generation sequencing (NGS) confirmation of the viral genome sequence.

## SARS-CoV-2 infection

Vero E6-TMPRSS2 and A549-hACE2 (RRID:CVCL_0023, ATCC) cell lines were seeded in 12-well plates at $3\times10^5$ cells/well. The next day, cells were infected at MOI 0.01 with an inoculation period of 1 hr, followed by a medium change to fresh culture medium. Supernatants were collected, and infected cells were fixed with 4% paraformaldehyde (PFA) for 30 min at indicated time before removal from the BSL-3 laboratory for fluorescence imaging using Cytation 5 (BioTek). Virus released into the supernatant was titrated using Vero cells via the limiting dilution method. Cell lines were confirmed to be mycoplasma-free using MycoStrip (InvivoGen, rep-mys-50).

## Acknowledgements

This work was supported by the Intramural Research Programs of NIBIB (ZIA EB000099-02), NHLBI, NCI, and NIAID at the National Institutes of Health (NIH). The contributions of the NIH authors are considered Works of the United States Government. The findings and conclusions presented in this paper are those of the authors and do not necessarily reflect the views of the NIH or the U.S. Department of Health and Human Services. We thank the Biophysics Resource in the Center for Structural Biology, Center for Cancer Research, NCI at Frederick for assistance with LC-MS studies. This work utilized the computational resources of the NIH HPC Biowulf cluster for molecular dynamics simulations.

## Additional information

### Funding

| Funder | Grant reference number | Author |
| --- | --- | --- |
| National Institutes of Health | ZIA EB000099-02 | Peter Schuck |

The funders had no role in study design, data collection and interpretation, or the decision to submit the work for publication.

## Author contributions

Huaying Zhao, Conceptualization, Resources, Data curation, Formal analysis, Supervision, Investigation, Methodology, Writing – original draft, Writing – review and editing; Tiansheng Li, Resources, Formal analysis, Investigation, Writing – review and editing; Sergio A Hassan, Resources, Data curation, Investigation, Visualization, Methodology, Writing – review and editing; Ai Nguyen, Resources, Validation, Investigation, Methodology; Siddhartha AK Datta, Resources, Formal analysis, Investigation, Methodology, Writing – review and editing; Guofeng Zhang, Investigation, Visualization, Methodology; Camden Trent, Investigation; Agata M Czaja, Investigation, Methodology; Di Wu, Resources, Investigation, Methodology; Maria A Aronova, Resources, Data curation, Formal analysis, Visualization, Methodology, Writing – review and editing; Kin Kui Lai, Resources, Investigation, Methodology, Writing – review and editing; Grzegorz Piszczek, Richard Leapman, Jonathan W Yewdell, Resources, Supervision, Methodology, Writing – review and editing; Peter Schuck, Conceptualization, Software, Supervision, Funding acquisition, Investigation, Visualization, Writing – original draft

## Author ORCIDs

Huaying Zhao ⓘD https://orcid.org/0000-0002-8827-6639
Grzegorz Piszczek ⓘD https://orcid.org/0000-0002-5270-3678
Peter Schuck ⓘD https://orcid.org/0000-0002-8859-6966

Reviewer #1 (Public review): https://doi.org/10.7554/eLife.108922.3.sa1
Reviewer #2 (Public review): https://doi.org/10.7554/eLife.108922.3.sa2
Reviewer #3 (Public review): https://doi.org/10.7554/eLife.108922.3.sa3
Author response https://doi.org/10.7554/eLife.108922.3.sa4

## Additional files

### Supplementary files

MDAR checklist

Supplementary file 1. Peptide and oligonucleotide sequences used in the present study.

### Data availability

Source data are available at the Harvard Dataverse https://doi.org/10.7910/DVN/4JPARC. Plasmids for N protein and mutants used in the biophysical studies will be shared on request. Raw data of all figures and SV-AUC analysis software can be accessed at the Harvard Dataverse (https://doi.org/10.7910/DVN/4JPARC).

The following dataset was generated:

| Author(s) | Year | Dataset title | Dataset URL | Database and Identifier |
|---|---|---|---|---|
| Zhao H, Li T, Hassan SA, Nguyen A, Datta SAK, Zhang G, Trent C, Czaja AM, Wu D, Aronova MA, Lai KK, Piszczek G, Leapman RD, Yewdell JW, Schuck P | 2025 | Replication Data for: Evolution of a fuzzy ribonucleoprotein complex in viral assembly | https://doi.org/10.7910/DVN/4JPARC | Harvard Dataverse, 10.7910/DVN/4JPARC |

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

## Appendix 1

### Estimation of the Impact of Surface Binding on Solution Concentrations in MP

Due to surface binding of particles, their concentration in the solution above the surface will drop. How much of an effect is this under our experimental conditions?

Let a volume $V$ of sample at a concentration $c_0$ be applied to a well on the coverslip with area $A$, of which a fraction $f_A$ is observed in the image. If we count $n$ total surface binding events visible in the camera, assuming no unbinding, the total number of surface-bound molecules is $n/f_A$ and the number of molecules left in solution is $c_0 V N_A - n/f_A$. This means the concentration in the solution volume is now $c_1 = \left(c_0 V N_A - n/f_A\right)\left(V N_A\right)^{-1} = c_0 - n/\left(f_A V N_A\right)$, or a loss of $\Delta c = n/\left(f_A V N_A\right)$ occurred.

In our experiments, the values are: Sample volume $V$ = 10 µL with an initial concentration $c_0$ = 300 nM (N monomer), i.e., we start with $\approx 1.8 \times 10^{12}$ molecules of N in solution. The well diameter is 3 mm; therefore, the well area $A \approx 7$ mm², and the observed spot size is $\approx 46$ µm², leading to the observed area fraction $f_A = 6.6 \times 10^{-6}$. Therefore, if we count $n$ = 10,000 particles (which exceeds the particle count in any of our experiments), the total number of particles absorbed would be $n/f_A$ = 1.5 × 10⁹ ($\approx$0.08% of the starting number), and accounting for (on average) approximately 10 copies of N per particle, this would amount to a reduction in concentration by approximately 0.8% or 2.5 nM. This is well within the concentration error.

This is in line with the observation that surface adsorption of proteins to glass is critical and needs to be prevented when working at picomolar concentrations (*Zhao et al., 2014*), but is ordinarily negligible when working at the mid nanomolar concentration range and non-functionalized surfaces.

It should be noted that this decrease in concentration is different from the decrease in count rate with time, which is governed by saturation of surface sites – an effect that diminishes the reduction of solute concentration above the surface with time. Finally, many particles exhibit desorption events, visible in a peak at negative contrast values, which will also further reduce the impact of surface adsorption on the solute concentration above the surface.

