## [Editor Report · eLife Assessment]

This is a **valuable** study that combines biophysical and evolutionary approaches to understand why particular mutations in the SARS-CoV-2 protein N arose during the COVID-19 pandemic. The evidence is **solid** and supports the conclusions.

---

## [Referee Report · Reviewer #1 (Public review)]

Summary:

The authors attempted to clarify the impact of N protein mutations on ribonucleoprotein (RNP) assembly and stability using analytical ultracentrifugation (AUC) and mass photometry (MP). These complementary approaches provide a more comprehensive understanding of the underlying processes. Both SV-AUC and MP results consistently showed enhanced RNP assembly and stability due to N protein mutations.

The overall research design appears well planned, and the experiments were carefully executed.

Strengths:

SV-AUC, performed at higher concentrations (3 µM), captured the hydrodynamic properties of bulk assembled complexes, while MP provided crucial information on dissociation rates and complex lifetimes at nanomolar concentrations. Together, the methods offered detailed insights into association states and dissociation kinetics across a broad concentration range. This represents a thorough application of solution physicochemistry.

Weaknesses:

Unlike AUC, MP observes only a part of solution. In MP, bound molecules are accumulated on the glass surface (not dissociated) thus concentration in solution should change as time develops. How does such concentration change impact the result shown here?

Comments on revisions:

The response from the authors is appropriate and reasonable.

---

## [Referee Report · Reviewer #2 (Public review)]

Summary:

In this manuscript, the authors apply a variety of biophysical and computational techniques to characterize the effects of mutations in the SARS-CoV-2 N protein on the formation of ribonucleoprotein particles (RNPs). They find convergent evolution in multiple repeated independent mutations strengthening binding interfaces, compensating for other mutations that reduce RNP stability but which enhance viral replication.

Strengths:

The authors assay the effects of a variety of mutations found in SARS-CoV-2 variants of concern using a variety of approaches, including biophysical characterization of assembly properties of RNPs, combined with computational prediction of the effects of mutations on molecular structures and interactions. The findings of the paper contribute to our increasing understanding of the principles driving viral self-assembly, and increases the foundation for potential future design of therapeutics such as assembly inhibitors.

Weaknesses:

For the most part, the paper is well-written, the data presented support the claims made, and the arguments made easy to follow. However, I believe that parts of the presentation could be substantially improved. I found portions of the text to be overly long and verbose and likely could be substantially edited; the use of acronyms and initialisms is pervasive, making parts of the exposition laborious to follow; and portions of the figures are too small and difficult to read/understand.

Comments on revisions:

The authors have adequately addressed all of my concerns.

---

## [Referee Report · Reviewer #3 (Public review)]

Summary:

This manuscript investigates how mutations in the SARS-CoV-2 nucleocapsid protein (N) alter ribonucleoprotein (RNP) assembly, stability, and viral fitness. The authors focus on mutations such as P13L, G214C, G215C combining biophysical assays (SV-AUC, mass photometry, CD spectroscopy, EM), VLP formation, and reverse genetics. They propose that SARS-CoV-2 exploits "fuzzy complex" principles, where distributed weak interfaces in disordered regions allow both stability and plasticity, with measurable consequences for viral replication.

Strengths:

* The paper demonstrates a comprehensive integration of structural biophysics, peptide/protein assays, VLP systems, and reverse genetics.

* Identification of both de novo (P13L) and stabilizing (G214C/G215C) interfaces provides a mechanistic insight into RNP formation.

* Strong application of the "fuzzy complex" framework to viral assembly, showing how weak/disordered interactions support evolvability, is a significant conceptual advance in viral capsid assembly.

* Overall, the study provides a mechanistic context for mutations that have arisen in major SARS-CoV-2 variants (Omicron, Delta, Lambda) and a mechanistic basis for how mutations influence phenotype via altered biomolecular interactions.

Weaknesses:

The weaknesses are shared via detailed comments to follow.

Comments on revisions:

The authors have addressed the criticisms of the original manuscript satisfactorily.

---

## [Author Response]

The following is the authors’ response to the original reviews.

**Reviewer #1 (Public Review):**
Summary:The authors attempted to clarify the impact of N protein mutations on ribonucleoprotein (RNP) assembly and stability using analytical ultracentrifugation (AUC) and mass photometry (MP). These complementary approaches provide a more comprehensive understanding of the underlying processes. Both SV-AUC and MP results consistently showed enhanced RNP assembly and stability due to N protein mutations.The overall research design appears well planned, and the experiments were carefully executed.Strengths:SV-AUC, performed at higher concentrations (3 µM), captured the hydrodynamic properties of bulk assembled complexes, while MP provided crucial information on dissociation rates and complex lifetimes at nanomolar concentrations. Together, the methods offered detailed insights into association states and dissociation kinetics across a broad concentration range. This represents a thorough application of solution physicochemistry.

We thank the Reviewer for this positive assessment.

Weaknesses:Unlike AUC, MP observes only a part of the solution. In MP, bound molecules are accumulated on the glass surface (not dissociated), thus the concentration in solution should change as time develops. How does such concentration change impact the result shown here?

We agree with the Reviewer that the concentration in solution above the surface will change with time; however, the impact of surface adsorption turns out to be negligible. To show this we have added a calculation as Supplementary Methods that is based on the number of imaged adsorption events, the fraction of imaged area to total surface area, and the initial sample volume and concentration. Under our experimental conditions the reduction is less than 1%, which is well within the range of experimental concentration errors.

This is in line with the observation that surface adsorption of proteins to glass is critical and needs to be prevented when working at picomolar concentrations (Zhao H, Mayer ML, Schuck P. 2014. Analysis of protein interactions with picomolar binding affinity by fluorescence-detected sedimentation velocity. Anal Chem 86:3181–3187. doi:10.1021/ac500093m), but is ordinarily negligible when working at the mid nanomolar concentration range. The difference in the MP experiments is that where usually the surface adsorption to glass and plastic is invisible, it is being imaged and quantified in MP. The negligible impact of surface adsorption on solution concentration in typical MP experiments is also in line with the results of several studies that have successfully measured dissociation constants of binding equilibria by MP (Young G et al., Science 360 (2018) 432; Wu & Piszczeck, Anal Biochem 592 (2020) 113575; Solterman et al. Angewandte Chemie 59 (2020) 10774) with samples in the 5-50 nM range and similar experimental setup. It should be noted that in the MP experiments no surface functionalization is employed, in contrast to optical biosensors that utilize surface-immobilized ligands and polymeric matrices and thereby enhance the surface binding capacity.

Even though this depletion effect is negligible under ordinary MP conditions, the Reviewer raises a good point and readers may have a similar question with this novel technique. For this reason, we have added in the MP section of the Methods the sentence “In either configuration, the impact of surface binding on the sample concentration is < 1% and negligible, as described in the Supplementary Methods S1.” and added the detailed calculations in the Supplement accordingly. The use of SV as a traditional, orthogonal technique and the observation of consistent results with those of MP should further dispel readers’ methodological concerns in this point.

**Reviewer #2 (Public Review):**
Summary:In this manuscript, the authors apply a variety of biophysical and computational techniques to characterize the effects of mutations in the SARS-CoV-2 N protein on the formation of ribonucleoprotein particles (RNPs). They find convergent evolution in multiple repeated independent mutations strengthening binding interfaces, compensating for other mutations that reduce RNP stability but which enhance viral replication.Strengths:The authors assay the effects of a variety of mutations found in SARS-CoV-2 variants of concern using a variety of approaches, including biophysical characterization of assembly properties of RNPs, combined with computational prediction of the effects of mutations on molecular structures and interactions. The findings of the paper contribute to our increasing understanding of the principles driving viral self-assembly, and increase the foundation for potential future design of therapeutics such as assembly inhibitors.

Thank you for highlighting the strengths of our paper and the potential impact on future design of therapeutics.

Weaknesses:For the most part, the paper is well-written, the data presented support the claims made, and the arguments are easy to follow. However, I believe that parts of the presentation could be substantially improved. I found portions of the text to be overly long and verbose and likely could be substantially edited; the use of acronyms and initialisms is pervasive, making parts of the exposition laborious to follow; and portions of the figures are too small and difficult to read/understand.

We are glad the Reviewer concurs the data support our conclusions, and finds the arguments easy to follow. We appreciate the comment that the work was not optimally presented. To address this point, we have identified multiple opportunities to streamline the text without jeopardizing the clarity. We have also rewritten the end of the Introduction.

As recommended, we have reduced and harmonized the use of acronyms and abbreviations throughout the text to improve readability. Specifically, we have now spelled out nucleic acid (NA), intrinsically disordered regions (IDR), full-length (FL), AlphaFold (AF3), and variants of concern (VOC).

Finally, we have improved the presentation of most figures, adding labels and new panels, and increased the label font sizes to facilitate more detailed inspections of the data.

**Reviewer #3 (Public Review):**
This manuscript investigates how mutations in the SARS-CoV-2 nucleocapsid protein (N) alter ribonucleoprotein (RNP) assembly, stability, and viral fitness. The authors focus on mutations such as P13L, G214C, and G215C, combining biophysical assays (SV-AUC, mass photometry, CD spectroscopy, EM), VLP formation, and reverse genetics. They propose that SARS-CoV-2 exploits "fuzzy complex" principles, where distributed weak interfaces in disordered regions allow both stability and plasticity, with measurable consequences for viral replication.Strengths:(1) The paper demonstrates a comprehensive integration of structural biophysics, peptide/protein assays, VLP systems, and reverse genetics.(2) Identification of both de novo (P13L) and stabilizing (G214C/G215C) interfaces provides a mechanistic insight into RNP formation.(3) Strong application of the "fuzzy complex" framework to viral assembly, showing how weak/disordered interactions support evolvability, is a significant conceptual advance in viral capsid assembly.(4) Overall, the study provides a mechanistic context for mutations that have arisen in major SARS-CoV-2 variants (Omicron, Delta, Lambda) and a mechanistic basis for how mutations influence phenotype via altered biomolecular interactions.

We are grateful for these comments highlighting this work as a significant conceptual advance.

Weaknesses:(1) The arrangement of N dimers around LRS helices is presented in Figure 1C, but the text concedes that "the arrangement sketched in Figure 1C is not unique" (lines 144-146) and that AF3 modeling attempts yielded "only inconsistent results" (line 149).The authors should therefore present the models more cautiously as hypotheses instead. Additional alternative arrangements should be included in the Supplementary Information, so the readers do not over-interpret a single schematic model.

We agree that in the absence of high-resolution structures the RNP models are hypothetical, and have now emphasized this in the Results, following the Reviewer’s recommendation. To present alternative arrangements that satisfy the biophysical constraints upfront, we have promoted the previous Supplementary Figure 11 showing different models to the first Supplementary Figure, and expanded it with examples of different oligomers. In this way it is referenced early on in the Results and in the legend to Figure 1C. We agree this strengthens the manuscript, as one of the take-home messages is the inherent polydispersity of the RNPs.

The fact that AF3 can only provide inconsistent results will not come as a surprise, given the substantial disordered regions of the complex, and is a drawback of AF3 rather than our structural model. We slightly emphasized this point so as to clarify that the presentation of the AF3-based RNP structure serves solely as supporting evidence that our hypothetical model is sterically reasonable.

The new Results paragraph reads:

“As suggested in the cartoon of Figure 1C, this supports the hypothesis of a three-dimensional arrangement with a central LRS oligomer with symmetry properties and dimensions similar to low resolution EM images of model RNPs (Carlson et al., 2022, 2020) and cryo-ET of RNPs in virions (Klein et al., 2020; Yao et al., 2020). It should be noted, however, that the arrangement sketched in Figure 1C is not unique and other subunit orientations could be envisioned that satisfy all constraints from experimentally observed binding interfaces, including different oligomers and anti-parallel subunits as illustrated in Supplementary Figure S1. Extending previous ColabFold structural predictions that show multiple N-protein dimers self-assembled via the LRS coiled-coils (Zhao et al., 2023), we attempted the AlphaFold modeling of RNPs combining multiple N dimers with SL7 RNA ligands, mimicking our biophysical assembly model. Current AlphaFold restrictions limit the prediction to pentamers of N-protein dimers with 10 copies of SL7 RNA. While only inconsistent results were obtained – which is not surprising given the large intrinsically disordered regions exceed the predictive power of AlphaFold – some models did produce an overall RNP organization similar to Figure 1C, suggesting such an arrangement is at least sterically reasonable with regard to possible N-protein subunit orientations in an RNP (Supplementary Figure S2)”

(2) Negative-stained EM fibrils (Figure 2A) and CD spectra (Figure 2B) are presented to argue that P13L promotes β-sheet self-association. However, the claim could benefit from more orthogonal validation of β-sheet self-association. Additional confirmation via FTIR spectra or ThT fluorescence could be used to further distinguish structured β-sheets from amorphous aggregation.

We completely agree that the application of multiple orthogonal biophysical methods can strengthen the conclusions. In addition to EM fibrils and CD spectra (a classical gold standard technique for protein secondary structure in solution), we already have support from ColabFold modeling, as well as NMR results from the Zweckstetter lab showing the potential for for β-sheet-like conformations.

Furthermore, we believe the evidence for the absence of ‘amorphous aggregates’ is very strong, as this would be inconsistent with the long-range order required to create the visibly fibrillar morphology in EM, and amorphous aggregates would be inconsistent with the increased solution viscosity. In this context, it is also highly relevant that the β-sheet-like secondary structure recorded by CD is concentration-dependent and reversible upon dilution. The long-range spatial order of fibrils is consistent with the formation of secondary structure in solution.

In addition, it must be kept in mind that what we see is specific to N-arm peptides carrying the P13L mutation (in EM, CD, and structural prediction) and does not occur in the other two N-arm peptides (ancestral N-arm and N-arm with deletion of 31-33), linker peptides, or C-arm peptides.

Most importantly, as elaborated in more detail below, we do not claim that fibril formation is physiologically relevant. At the heart of this – in the context of the evolution of fuzzy complexes – is that the P13L mutation creates additional weak protein-protein interactions. Indeed, the assembly of fibrils geometrically requires at least two interfaces for each subunit. These weak interactions are at play physiologically in the context of the disordered RNP particles, and in macromolecular condensates, but not in the formation of fibrils. Therefore, while we appreciate the suggestion for FTIR spectra ThT staining, we are afraid further emphasis on the fibril structure might confuse the reader, and therefore we would rather clarify upfront that these fibrillar assemblies are not thought to form in vivo from full-length protein, but merely demonstrate the presence of N-arm self-association interfaces in the model of truncated peptides.

Accordingly, we have amended the Results paragraph reporting the fibrils:

“Thus, the N-arm mutation P13L is responsible for the formation of fibrils in N-arm peptides after prolonged storage. Some of these N-arm fibrils exhibit a twisted morphology with width of ≈5 nm (Figure 2A), in some instances exhibiting patterns of strand breaks. Such fibrils are frequently encountered in proteins that can stack β-sheets, such as in amyloids (Paravastu et al., 2008). While we have not observed fibril formation in the context of full-length N, and have no evidence such fibrils are physiologically relevant, their occurrence in solutions of truncated N-arm peptide nonetheless demonstrates the introduction of ordered N-arm self-association interfaces in conformations of P13L mutants.”

And more completely summarized experimental evidence prior to describing the ColabFold prediction results (which previously did not include mention of the NMR):

“Finally, confirming the interpretation of the EM images and the CD data, as well as the b-structure propensity reported from NMR data (Zachrdla et al., 2022), the structural prediction of N[10-20]:P13L in ColabFold displayed oligomers with stacking b-sheets …”

(3) In the main text, the authors alternate between emphasizing non-covalent effects ("a major effect of the cysteines already arises in reduced conditions without any covalent bonds," line 576) and highlighting "oxidized tetrameric N-proteins of N:G214C and N:G215C can be incorporated into RNPs". Therefore, the biological relevance of disulfide redox chemistry in viral assembly in vivo remains unclear. Discussing cellular redox plausibility and whether the authors' oxidizing conditions are meant as a mechanistic stress test rather than physiological mimicry could improve the interpretation of these results.The paper could benefit if the authors provide a summary figure or table contrasting reduced vs. oxidized conditions for G214C/G215C mutants (self-association, oligomerization state, RNP stability). Explicitly discuss whether disulfides are likely to form in infected cells.

We thank the Reviewer for raising this most interesting point. The reason why the biological relevance of N dilsulfides remains unclear is simply that this is still unknown, unfortunately. Recently, Kubinski et al. have strongly argued for the formation of disulfides in infected cells, but in our view the evidence remains weak since the majority of disulfide bonds in that work presented as post-lysis artifacts, and it appears the non-covalent effects alone could explain the physiological observations. We aimed for a balanced presentation and wrote in the relevant Results section:

“Covalent disulfide bonds in the LRS in non-reducing conditions were found to further promote LRS oligomerization. However, there is no conclusive data yet whether covalent bonds in the LRS occur in vivo, or any G215C effect is entirely non-covalent due to the significant strengthening of LRS helix oligomerization (see Discussion).”

Despite the uncertainty regarding physiological disulfide bond formation, we believe it is useful to ask whether covalently crosslinked N dimers would aid or constrain RNP assembly in our biophysical model. We have now better explained this motivation in the Results section describing the RNP experiments:

“Even though it is still unclear whether disulfide bonds of N cysteine mutants form in vivo, we were curious about the impact of disulfide-linked oligomers of the cysteine mutants on their RNP structure and stability in our biophysical assembly model.”

The referenced paragraph from the Discussion reads:

“Regarding the cysteine mutations that have been repeatedly introduced in the LRS prior to the rise of the Omicron VOCs, it is an open question whether they lead to covalent bonds in vivo or in the VLP assay. While examples of disulfide-linked viral nucleocapsid proteins have been reported (Kubinski et al., 2024; Prokudina et al., 2004; Wootton and Yoo, 2003), a methodological difficulty in their detection is artifactual disulfide bond formation post-lysis of infected cells (Kubinski et al., 2024; Wootton and Yoo, 2003). However, our results clearly show that a major effect of the cysteines already arises in reduced conditions without any covalent bonds, through extension of the LRS helices, and concomitant redirection of the disordered N-terminal sequence. While oxidized tetrameric N-proteins of N:G214C and N:G215C can be incorporated into RNPs, the covalent bonds provided only marginally improved RNP stability. Interestingly, the introduction of cysteines imposes preferences of RNP oligomeric states dependent on oxidation state, consistent with our MD simulations highlighting the impact of cysteine orientation of 214C versus 215C relative to the hydrophobic surface of the LRS helices. Overall, considering potentially detrimental structural constraints from covalent bonds on LRS clusters seeding RNPs, energetic penalties on RNP disassembly, as well as the required monomeric state of the LRS helix for interaction with the NSP3 Ubl domain (Bessa et al., 2022), at present it is unclear to what extent the formation of disulfide linkages between LRS helices would be beneficial or detrimental in the viral life cycle.”

We feel that this text addresses the Reviewer’s comment, and that expanding the existing discussion further would conflict with other recommendations to shorten and focus the text.

Finally, we have addressed the valuable suggestion of a new table summarizing the oligomeric state and self-association of the different cysteine mutants by inserting a new column in the existing Table 1 reporting all species’ oligomeric state at low micromolar concentrations. In this way they can be compared at a glance with the other mutants as well. A more detailed comparison of the concentration-dependent size-distribution is provided in Figure 4.

(4) VLP assays (Figure 7) show little enhancement for P13L or G215C alone, whereas Figure 8 shows that P13L provides clear fitness advantages. This discrepancy is acknowledged but not reconciled with any mechanistic or systematic rationale. The authors should consider emphasizing the limitations of VLP assays and the sources of the discrepancy with respect to Figure 8.

We thank the Reviewer for this comment, which highlights a very important point.

For clarification and to improve the cohesion of the manuscript we have inserted a reference to the Discussion after the presentation of the VLP results, which provides a natural transition to the following description of the reverse genetics experiments:

“As expanded on in the Discussion, the failure to observe enhancement by P13L alone may be related to limitations of the VLP assay in sensitivity, including the restriction to a single round of infection, and protein expression levels.”

This references a paragraph in the Discussion about the limitations of the VLP assay in general and the reasons we believe the enhancement by P13L alone was not picked up:

“…While this assay has been widely used for rapid assessment of spike protein and N variants (Syed et al., 2021), it has limitations due to the addition of non-genomic RNA and the lack of double membrane vesicles from which gRNA emerges through the NSP3/NSP4 pore complex potentially poised for packaging (Bessa et al., 2022; Ke et al., 2024; Ni et al., 2023). It should also be recognized that the results do not directly reflect the relative efficiency of RNP assembly only, since protein expression levels, their localization, and their posttranslational modifications are not controlled for. Susceptibility for such factors might be exacerbated with mutations that modulate weak protein interactions. For example, as shown previously (Syed et al., 2024; Zhao et al., 2024), a GSK3 inhibitor inhibiting N-protein phosphorylation significantly enhances VLP formation and eliminates the advantage provided for by the N:G215C mutation relative to the ancestral N – presumably due to an increase in assembly-competent, non-phosphorylated N-protein erasing an affinity advantage. A similar process may be underlying the absent or marginal improvement in VLP readout from the cysteine LRS mutants and P13L at the achieved transfection level in the present work, and the enhanced signal from R203K/G204R and R203M the latter being consistent with previous reports (Li et al., 2025; Syed et al., 2021) modulating protein phosphorylation. Nonetheless, mirroring the results of the biophysical in vitro experiments, the addition of RNP-stabilizing P13L and G214C mutations on top of R203K/G204R led to a significantly larger VLP signal.

The VLP assay may be limited in sensitivity to mutation effects due to its restriction to a single round of infection. To avoid this and other potential limitations of the VLP assay for the study of viral packaging, for the key mutation N:P13L we carried out reverse genetics experiments. These showed the sole N:P13L mutation significantly increases viral fitness (Figure 8).”

(5) Figures 5 and 6 are dense, and the several overlays make it hard to read. The authors should consider picking the most extreme results to make a point in the main Figure 5 and move the other overlays to the Supplementary. Additionally, annotating MP peaks directly with "2×, 4×, 6× subunits" can help non-experts.

We completely agree with the Reviewer – these figures were very dense. To mitigate this problem without having the reader to switch back-and-forth to the supplement, we subdivided the panels of Figure 5 and showed only a subset of curves in each. In this way the data are easier to read while still readily compared. It is a large figure, but it contains the key data for the present work and is therefore worthwhile to have in one place. For the MP histogram data we also have inserted the suggested peak labels. Similarly, we have split Figure 6A into two panels for clarity.

(6) The paper has several names and shorthand notations for the mutants, making it hard to keep up. The authors could include a table that contains mutation keys, with each shorthand (Ancestral, Nο/No, Nλ, etc.) mapped onto exact N mutations (P13L, Δ31-33, R203K/G204R, G214C/G215C, etc.). They could then use the same glyphs (Latin vs Greek) consistently in text and figure labels.

Yes, we agree this is a problem and we apologize for the confusion. However, it is not possible to refer exclusively to either Latin or Greek terminology, which we feel would be even more detrimental to readability (the former being exhaustively lengthy and the latter being imprecise). But we have used a rational system: If the complete set of mutations of a variant are present, then its Greek letter will be used as an abbreviation, and otherwise we use Latin amino acid/position indicators for individual mutations or combinations thereof. Unfortunately, previously we inadvertently failed to explicitly mention this, and we are most grateful for the Reviewer to point this out.

We have now rectified this by including upfront the sentence:

“We will adopt a nomenclature where the complete set of defining mutations of a variant will be referred to by its Greek letter, i.e., N:P13L/R203K/G204R/G214C is N_­­λ_, and analogously the set of Omicron mutations N:P13L/Δ31-33/R203K/G204R are referred to as N_ο_; see Table 1”

This will define the two shorthands N_λ_ and N_ο_ used. Furthermore, as suggested and pointed to in the text, Table 1 does provide the keys to mutation and variants, including the information in which variant any of the other mutations studied here occur.

(7) The EM fibrils (Figure 2A) and CD spectra (Figure 2B) were collected at mM peptide concentrations. These are far above physiological levels and may encourage non-specific aggregation. Similarly, the authors mention" ultra-weak binding energies that require mM concentrations to significantly populate oligomers". On the other hand, the experiments with full-length protein were performed at concentrations closer to biologically relevant concentrations in the micromolar range. While I appreciate the need to work at high concentrations to detect weak interactions, this raises questions about physiological relevance.

This is indeed an important point to clarify. We agree that much lower nucleocapsid protein concentrations are present in the cytosol on average, and these were used in our RNP assembly experiments. However, there are at least two important physiologically relevant cases where high local N concentrations do occur:

(1) Once assembled in RNPs, the disordered N-terminal extensions are locally at a very high concentration within the volume they can explore while tethered to the NTD. A back-of-the-envelope calculation assuming 12 N-protein subunits confining 12 N-terminal extensions to the volume of a single RNP (≈14x14x14 nm^3^ by cryoEM; Klein et al 2020) leads to an effective concentration of 7.4 mM. Obviously the N-arm peptides are not completely free and there will be constraints that would hinder or promote encounter complex probability, but interfaces with mM Kd are clearly strong enough to populate Narm-Narm contacts extending from N-protein in the RNP.

Additionally, any interaction where N-proteins are brought in close proximity could allow weak N-arm interactions to provide additional stability. Besides the RNP, we demonstrate this in our Results for nucleic-acid liganded N tetramers (Figure 4B), but this might similarly occur in complexes with NSP3 or host proteins. Generally, it is quite common that small additional binding energies play important roles in the modulation of multivalent protein complexes.

(2) Within the macromolecular condensate the local concentration will be substantially higher than on average within the infected cell. While we do not know its precise concentration, it is well-established that the sum of many ultra-weak interactions is driving the formation of this dense liquid phase. In our previous eLife paper (Nguyen et al., 2024) we have shown LLPS is suppressed with the R203K/G204R mutation, but it is ‘rescued’ with the additional P13L/del31-33 mutation of the Omicron variant showing strong LLPS. Similarly, LLPS is suppressed by the LRS mutant L222P, but rescued in conjunction with P13L. This is another biologically relevant scenario where weak interactions are critical.

We have emphasized these points in the revised manuscript as described below.

Specifically:(a) Could some of the fibril/β-sheet features attributed to P13L (Figure 2A-C) reflect non-specific aggregation at high concentrations rather than bona fide self-association motifs that could play out in biologically relevant scenarios?

We understand this concern from the experience with proteins that often have limited solubility and tendencies to aggregate, sometimes accompanied by unfolding and driven by hydrophobic interactions, or clustering on the path to LLPS. However, we are struggling to reconcile the picture of non-specific aggregation with the context of our P13L N-arm peptides. The term ‘non-specific aggregation’ implies the idea of amorphous aggregates, which we would contend is inconsistent with the observed geometry of fibrils, which exhibit long-range order. In addition, non-specific aggregation does not lead to increased solution viscosity, which we describe, but fibril formation does. Another connotation of ‘aggregates’ is irreversibility. However, we find the beta-sheet-like conformation seen at 1 mM becomes significantly more disordered when the same sample is diluted to 0.4 mM peptide. This is consistent with a reversible self-association driven by a conformational change toward ordered secondary structure.

To highlight the reversibility, we have clarified the description: “Interestingly, diluting the 1 mM sample (solid) to a concentration of 0.4 mM (dashed) reveals a large shift in the far-UV spectra … both indicative of a significant increase of disorder upon dilution. This is consistent with the stabilization of b-sheets in a reversible, strongly cooperative self-association process with an effective K_D_ in the high μM to low mM range.”

We have also inserted a concentration conversion to mg/ml units, which shows even 1 mM of peptides is only ≈5 mg/ml, i.e. not excessively high. “While the ancestral N-arm at ≈1 mM (≈ 4.6 mg/ml) concentrations exhibits CD spectra with a minimum at ≈200 nm typical of disordered conformations (black)”

With regard to the question of specificity, we have studied similar N-arm peptides without P13L mutations and with the 31-33 deletion under equivalent conditions. But we observe the reversible self-association, conformational change, and fibril formation only for those containing the P13L mutation, consistent with ColabFold predictions. Neither did we observe fibrils with disordered C-arm peptides.

How these weak self-association motifs in the N-arm can be physiologically relevant in the context of full-length protein modulating the stability of multi-molecular complexes and enhancing LLPS was outlined above, and further clarified in the manuscript as detailed below.

(b) How do the authors justify extrapolating from the mM-range peptide behaviors to the crowded but far lower effective concentrations in cells?

As pointed out above, the key to this question is the local preconcentration as the N-arm peptides are tethered to the rest of protein in the context of flexible multi-molecular assemblies. Another mechanism to consider is the formation of condensates. The response to the next comment will expand on this.

The authors should consider adding a dedicated section (either in Methods or Discussion) justifying the use of high concentrations, with estimation of local concentrations in RNPs and how they compare to the in vitro ranges used here. For concentration-dependent phenomena discussed here, it is vital to ensure that the findings are not artefacts of non-physiological peptide aggregation..

The use of high concentration in biophysical experiments is quite common, for example, in NMR or crystallography, insofar as they elucidate molecular properties. We believe this is obvious; the Reviewer will certainly agree with us, and this does not require further elaboration. The property observed in this case is the existence of specific, weak protein self-association interfaces in the N-arm.

Our response to the Reviewer’s point 7(a) addresses the distinction between artefactual aggregation and self-association of N-arm peptides. The relevance of these weak protein self-association interfaces in the context of the full-length protein is the second underlying question.

As we have previously stated in a dedicated Results paragraph:

“In contrast to the modulation of the coiled-coil LRS interfaces, the de novo creation of the N-arm self-association interface through beta-sheet interactions enabled by P13L cannot be readily observed in full-length N-protein at low M concentrations. Similar to the ancestral LRS interface, it provides only ultra-weak binding energies that require mM concentrations to significantly populate oligomers. This is fully consistent with the previous observation by SV-AUC that neither N:P13L,31-33 nor N_o_ with the full set of Omicron mutations show any significant higher-order self-association at low M concentrations, whereas at high local concentrations – as observed in phase-separated droplets – they can modulate and cooperatively enhance self-association processes (Nguyen et al., 2024). (If fact, P13L can substitute for the LRS promoting LLPS, as observed in the rescue of LLPS by N:P13L,31-33/L222P mutants whereas N:L222P LRS-abrogating mutants are deficient in LLPS.) Another process that increases the local concentration of N-arm chains is the tetramerization of full-length N-protein. As described earlier, occupancy of the NA-binding site in the NTD allosterically promotes self-assembly of the LRS into higher oligomers (Zhao et al., 2021). We hypothesized that these oligomers may be cooperatively stabilized by additional N-arm interactions in P13L mutants.”

To state completely unambiguously why weak interfaces are important, we have followed the Reviewer’s suggestion and added an additional clarification already earlier, at the end of the P13L Results section:

“While this self-association interface in the P13L N-arm is weak and its direct observation in biophysical experiments requires mM concentrations, which far exceed average intracellular concentration of N, such weak interactions can become highly relevant physiologically when high local concentrations are prevailing, for example, when the disordered extension is preconcentrated while tethered within macromolecular assemblies as in the RNP, or in macromolecular condensates.”

Furthermore, we have added early in the Discussion:

“Even though the solution affinity of the N-arm P13L interface is ultra-weak, the average local concentration of N-arm chains across the RNP volume (in a back-of-the-envelope calculation assuming a ≈14 nm cube (Klein et al., 2020) with a dodecameric N cluster) is ≈7.4 mM, such that disordered N-arm peptides could well create populations of N-arm clusters stabilizing RNPs through this interface. However, besides the RNP-stabilizing mutants we have also observed unexpected RNP destabilization by the ubiquitous R203K/G204R double mutation, which may be caused by the introduction of additional charges close to the self-association interface in the LRS. In our experiments, this destabilization is more than compensated for by the P13L mutation. (Another scenario where ultra-weak interactions can have a critical impact is in molecular condensates. We previously reported the suppression of LLPS by the R203K/G204R mutation, which is rescued by the additional P13L/Δ31-33 mutation (Nguyen et al., 2024). This is consistent with compensatory weak stabilizing and destabilizing impacts of weak interactions on the RNP observed here.)”

**Reviewer #1 (Recommendations for the Authors):**
In Figure 1B, it is unclear what the orange lines connecting polypeptides represent, as well as the zig-zag orange lines in the N-arm.

We thank the Reviewer for this comment. We intended this to represent regions of self-association but recognize the patterned background is confusing. We have changed this now to solid-colored backgrounds, and indicated this in the figure legend:

“Regions of self-association are indicated by shaded backgrounds.”

Regarding presentation, in Figure 5 (MP), the relationship between mass and oligomer size should be shown more clearly.

We agree. To this end we have labeled the peaks in the MP histograms in Figure 5 with the oligomeric state of the 2N/2SL7 subunits.

**Reviewer #2 (Recommendations for the Authors):**
I find the science of the paper to be convincing and compellingly supported.

Thank you for this positive statement.

My primary complaints are with presentation or minor technical questions that, honestly, primarily arise due to my own ignorance and unfamiliarity with some of the techniques employed.My primary issue is with the figures. I find, generally, the text in axes labels, ticks, and legends to be too small to comfortably read. This is particularly true in the CD spectra andother data presented in Figures 1D, 2B, 4, 5, 6, and 8.

We agree and have increased the font size of all text and labels of the plots in Figure 1, 2, 4, 5, 6, and 8.

I also found the use of initialisms to be a bit overbearing and inconsistent. For example, the authors repeatedly switch between spelling out "nucleic acid" and the initialism "NA" (which is also never explicitly spelled out in the text). With the already substantial length of the text, my own personal opinion would be to suggest spelling out all initialisms in the interest of making the reading easier.

This is a valid criticism. To improve the readability, we have followed this advice and systematically spelled out “nucleic acid” instead of using “NA”. Similarly, we have now written out full-length instead of the abbreviation FL, and omitted the abbreviation IDR for intrinsically disordered regions, as well as VOC for variant of concern, and AF3 for AlphaFold.

Regarding the reference to mutants, we have now explained upfront the system of Latin and Greek nomenclature we consistently applied.

“We will adopt a nomenclature where the complete set of defining mutations of a variant will be referred to by its Greek letter, i.e., N:P13L/R203K/G204R/G214C is N­­_l_, and analogously the set of Omicron mutations N:P13L/Δ31-33/R203K/G204R are referred to as N_ο_; see Table 1”

I found the text to be verbose, bordering on overly so; the Introduction is more than two pages long. The section "Enhanced oligomerization of the leucine-rich sequence through cysteine mutations" has two long paragraphs of introduction before the present results are discussed, et cetera. An (admittedly, very rough) estimation of the length of the paper places it at ~9,000 -10,000 words long, and I think that the presentation might benefit from significant editing andshortening.

We agree the manuscript is longer than would be desirable, and we generally prefer not to insert mini-introductions into Results sections. On the other hand, in order to make a solid contribution to understanding the big picture of fuzzy complexes in molecular evolution of RNA virus proteins it is indispensable to go into the details of RNP assembly and several of the interfaces. Therefore, we feel the length is in the range that it needs to be without losing clarity. In addition, other Reviewer suggestions to extend the discussion, for example, of limitations of VLP assays and the in vivo state of cysteines, conflict with significant shortening.

In the particular case of the cysteine mutations, cited by the Reviewer, we believe it is important to add detailed background on G215C, because the Results proceed in a comparison of the self-association mode between G215C and G214C. This is of significant interest in the present context not only for the independent introduction of interface-enhancing mutations highlighting the evolution of fuzzy complexes, but also because it illustrates the pleomorphic ability of RNPs.

Nonetheless, we have slightly shortened this text and merged the background into a single paragraph. More generally, we have critically reread the text to remove tangential sentences where possible and to make it more concise.

I have a few more specific comments.In Figure 1A, I suggest explicitly labeling the location of the LRS, as it comes up repeatedly.

Yes, we thank the Reviewer for this suggestion and have introduced this label in Figure 1A.

In Figure 1B, the legend indicates that the red lines indicate "new inter-dimer interactions." However, these red lines are overlayed on a vertical stripe of red squiggles; it is unclear to me and not explicitly described in the legend what these squiggles are meant to illustrate.

We agree this background was confusing. As mentioned in our Response to Reviewer #1 we have replaced the structured background with a solid background and explained in the figure legend that these areas depict regions of self-association.

On lines 44-45, the authors state, "The IDRs amount to 45%, ..." 45% of what?

Thank you, this was unclear. We have now clarified “The IDRs amount to ≈45% of total residues”

In lines 244 - 246, the authors compare the sizes of complexes in reducing versus non- reducing conditions as measured by dynamic light scattering, stating, "However, dynamic light scattering (DLS) revealed the presence of N210-246:G214C complexes with hydrodynamic radii 244 ranging from 6 to 40 nm (in comparison to 1-2 nm for N210- 246:G215C(Zhao et al., 2022)) in reducing conditions, and slightly larger in non-reducing conditions (Supplementary Figure S4)." Using this single statistic seems to me to be a less-than-ideal way of characterizing what seems to me to be happening here. In Supplementary Figure 4, it appears to me that what is happening is that in non-reduced conditions, the sample is monodisperse, whereas in reducing conditions, the distribution becomes polydisperse/bimodal, with two clearly separate populations. I feel that this could use a morethorough description rather than just stating the overall range of particle sizes.

Yes, the Reviewer is correct – it is indeed a good idea to be more precise here. To this end we have carried out cumulant analyses on the autocorrelation functions, as a time-honored method to quantify the polydispersity. Both samples are polydisperse, but more so in reducing conditions. We have now added “For N210-246:G214C a cumulant analysis results in radii of 8.8 nm and 10.6 nm and polydispersity indices of 0.40 and 0.35 for reducing and non-reducing conditions, respectively”

Finally, I have one remaining comment that is a result of my own inexperience with circular dichroism and interpreting the spectra. For me personally, I would appreciate a more thoroughdescription/illustration of the statistics involved in the CD spectra, but perhaps this is not necessary for people who are more familiar with interpreting these kinds of data. For example, in Figure 1D, it is not clear to me what the error bars/confidence intervals for the CD data look like. I see many squiggles, some of which the authors claim are significant (e.g., the differences between ~215 - 230 nm), and others are not worthy of comment. Let's say, for example, that I fit a smoothed spline through these data and then measure the magnitude of the fluctuations from that spline to define/quantify confidence intervals. What does that distribution look like? Or maybe the confidence intervals are so small that all squiggles are significant?

Thank you, this is a good question. As mentioned in the methods section, the CD spectra shown are averages of triplicate scans. Therefore, it is straightforward to extract the standard deviation at each wavelength from the three measurements (although a spline would probably work just as well). The values are what one would expect for the squiggles to be random noise. In the region 215 – 220 nm characteristic for helical secondary structure the standard deviations are small relative to the separation between curves, which indicates that the differences are highly significant. Naturally, the curves do overlap in other spectral regions, which would make a plot including the wavelength-dependent error bars or confidence bands too crowded. Therefore, we have kept the plot of the averaged triplicate scans, but have now provided the average standard deviations for all species in the figure legend and mentioned their significant separation:

“Triplicate scans yield average standard deviations of 0.13 (N), 0.17 (N+SL7), 0.16 (N_l_), and 0.21 (N_l_ +SL7) 10^3^ deg cm^2^/dmol, respectively, with non-overlapping confidence bands for the different species, for example, between 215-220 nm.”

**Reviewer #3 (Recommendations for the Authors):**
(1) The Discussion reiterates much of the background (mutational tolerance, fuzziness, SLiMs) already covered in the Introduction, diluting focus on the key new findings. The authors should consider shortening and refocusing the discussion on the main contributions in light of existing knowledge of viral assembly.

In the Introduction we have provided background on intrinsically disordered proteins in general and their mutational tolerance, as well as the concept of fuzzy complexes. The first several paragraphs of the Discussion have a different focus, which is protein binding interfaces between viral proteins (obviously key in fuzzy complexes), specifically their modulation and the remarkable de novo introduction of binding interfaces. We believe this deserves emphasis, since this highlights a novel aspect of fuzziness, for the mutant spectrum of RNA viruses to encode a range and of assembly stabilities and architectures.

To reduce redundancy between the end of the Introduction and the beginning of the Discussion, we have shortened the last paragraph of the Introduction and removed its preview of the conclusions, as described in the response to the next comment of the Reviewer (see below).

Unfortunately, the length of the Discussion is dictated in part also by the need to discuss methodological aspects, among them the limitations of VLP assays, and the redox state of the cysteine in the LRS mutants, which were important points recommended by other suggestions of the Reviewers. Similarly, we believe the discussion of other potential functions of Omicron N-arm mutations is warranted, as well as the background of the R203K/G204R double mutation that has attracted significant attention in the field due to its effects on phosphorylation and expression of truncated N species that also form RNPs. Our goal was to integrate the results by us and other laboratories regarding specific mutation effects into a comprehensive picture of molecular evolution of N, which we believe the framework of fuzzy complexes can provide.

(2) The Abstract and early Introduction set a broad stage (IDPs, fuzziness), but don't explicitly state the concrete hypotheses that the experiments test. Please add 2-3 sentences in the Introduction that enumerate testable hypotheses, e.g.:(a) P13L creates a new N-arm interface that increases RNP stability.(b) G214C/G215C strengthens LRS oligomerization to stabilize higher-order N assemblies.

We agree the introduction can be improved. However, it seems to us that it cannot be neatly framed in the hypothesis – answer dichotomy, without losing a lot of nuances and without requiring an even longer and more detailed introduction.

One of the main questions is to test whether the framework of fuzzy complexes can be applied to understand molecular evolution of N, and we feel the introduction is already flowing well towards this:

“ … In fuzzy complexes the total binding energy is distributed into multiple distinct ultra-weak interaction sites (Olsen et al., 2017). Similar to individual RNA virus proteins with loose or absent structure, maintaining disorder and a spatial distribution of low-energy interactions in the protein complexes may increase the tolerance for mutations and improve evolvability of protein complexes.\

The unprecedented worldwide sequencing effort of SARS-CoV-2 genomes during its rapid evolution in humans provides a unique opportunity to examine these concepts. ...”

To bring this to a more concrete set of questions in the end, we have shortened and rewritten the last paragraph in the Introduction:

“To examine how architecture and energetics of RNP assemblies can be impacted by N-protein mutations we study a panel of N-proteins derived from ancestral Wuhan-Hu-1 and different VOCs, including Alpha, Delta, Lambda, and Omicron (see Table 1), in biophysical experiments, VLP assays, and mutant virus. Specifically, we ask how the RNP size distribution and life-time is modulated by: (1) the novel binding interface created by the P13L mutation of Omicron; (2) enhancements of other weak self-association interfaces through G215C of Delta and G214C of Lambda; (3) the ubiquitous R203K/G204R double mutation of Alpha, Lambda, and Omicron. We also test whether the P13L mutation improves viral fitness, similar to G215C and R203K/G204R. The results are discussed in the framework of fuzzy complexes and molecular evolution of N in the course of viral adaptation to the human host. Understanding the salient features of the binding interfaces in viral assembly and their evolution expands our foundation for the design of therapeutics such as assembly inhibitors.”